



# Recent glacier mass balance and area changes in the Kangri Karpo Mountain derived from multi-sources of DEMs and glacier inventories

Wu Kunpeng [1, 2], Liu Shiyin [2, 3*], Jiang Zongli [4], Xu Junli [5], Wei Junfeng [4], Guo Wanqin [2]

[1]School of Resources and Environment, Anqing Normal University, Anqing, Anhui, China

[2]State key Laboratory of Cryospheric Sciences, Northwest Institute of Eco-Environment and Resources, Chinese Academy of Sciences, Lanzhou, China

[3]Institute of International Rivers and Eco-Security, Yunnan University, Yunnan, China

[4]Department of Geography, Hunan University of Science and Technology, Xiangtan, China

[5]Department of Surveying and Mapping, Yancheng Teachers University, Yancheng, China

Correspondence to LIU Shiyin at liusy@lzb.ac.cn or WU Kunpeng at wukunpeng2008@lzb.ac.cn

**Abstract.** Due to the effect of Indian monsoon, the Kangri Karpo Mountain, located in southeast Tibetan Plateau, is the most humid region of Tibetan Plateau, and become one of the most important and concentrated regions with maritime (temperate) glaciers development. Glacier mass loss in Kangri Karpo Mountain is important contributor to global mean sea level rise, and it change runoff distribution, increase risk of glacial lake outburst floods (GLOFs). Because of their difficult accessibility and high labor costs, the knowledge of glaciological parameters of glaciers in the Kangri Karpo Mountain is still limited. This study presents glacier elevation changes in the Kangri Karpo Mountain, by utilizing geodetic methods based on digital elevation models (DEM) derived from Topographic Maps (1980), the Shuttle Radar Topography Mission (SRTM) DEM (2000), and TerraSAR-X/TanDEM-X (2014). Glacier area and length changes were derived from Topographical Maps and Landsat TM/ETM+/OLI images between 1980 and 2015. Our results show that the Kangri Karpo Mountain contains 1166 glaciers, with an area of 2048.50 $\pm$48.65 km$^2$ in 2015. Ice cover in the Kangri Karpo Mountain diminished by 679.51 $\pm$ 59.49 km$^2$ (24.9% $\pm$ 2.2%) or 0.71% $\pm$0.06% a$^{-1}$ from 1980–2015, however, with nine glaciers in advance from 1980–2015. Glaciers with area of 788.28 km$^2$ in the region, as derived from DEM differencing, have experienced a mean mass deficit of 0.46 $\pm$0.08 m w.e. a$^{-1}$ from 1980-2014. These glaciers showed slight accelerated shrinkage and significant accelerated mass loss during 2000–2015 compared to that during 1980–2000, which is consistent with the tendency of climate warming.

## 1    Introduction

Glaciers in the Tibetan Plateau (TP), as a key components in cryosphere system (Li et al., 2008), are the water resource of many major rivers and lakes (Immerzeel et al., 2010). Glacier mass balance is a valuable indicator to understand the climate variability on the TP (Oerlemans, 1994; Yao et al., 2012a). Under the background of global warming, many mountain glaciers have progressively shrunk in mass and extent in past decades (IPCC, 2013). However, glaciers with positive mass balance have also been reported in recent years, especially in the central Karakoram, eastern Pamir and the western TP (Bao et al., 2015; Gardelle et al., 2012b; Gardelle et al., 2013; Kääb et al., 2015; Neckel et al., 2014; Yao et al., 2012a). The relationship between glacier mass



balance and climate change, and the knowledge of water source in cryosphere and its disaster risks,
are become an advanced research hotspot.
Glaciers in the Kangri Karpo Mountain is known as temperate characteristic under a warming
climate and abundant precipitation from Indian Monsoon (Li et al., 1986; Shi and Liu, 2000).
Literature review shows the lack of studies of glacier changes in the Kangri Karpo Mountain on a
longer time scale, especially the studies of mass balance. Based on digitized glacier inventories
from Topographic Maps and remote sensing images, or in situ measurements, glaciers in the
Kangri Karpo Mountains have experienced an intense area reduction, mass deficit, and continued
terminus retreat (Li et al., 2014; Yang et al., 2010; Yang et al., 2008; Yao et al., 2012a). While
some previous studies showed that the phenomenon of glacier advance exists in the Kangri Karpo
Mountain. Based on aerial photographs, China – Brazil Earth Resources Satellite (CBERS) image
and Landsat Thematic Mapper (TM) image, about 60% of glaciers in the region have been losing
their mass and other glaciers have advanced during 1980 to 2001 (Liu et al., 2006). Shi et al.
(2006) attributed the complex behavior of glacier dynamics to the increase of precipitation
suppressing glacier melting.
Although glacier mass deficit was established, the results did differ from each other (Gardelle
et al., 2013; Gardner et al., 2013; Kääb et al., 2015; Neckel et al., 2014). Using SRTM DEM and
SPOT5 DEM (24 November 2011), glaciers experienced a mean thinning of $0.39 \pm 0.16$ m a$^{-1}$ in
eastern Nyainqentanglha Mountains (Gardelle et al., 2013). Based on ICESat and SRTM, (Kääb et
al., 2015), (Neckel et al., 2014) and (Gardner et al., 2013) acquired different results over eastern
Nyainqentanglha Mountain, with glacier thickness loss of $1.34 \pm 0.29$ m a$^{-1}$, $0.81 \pm 0.32$ m a$^{-1}$ and
$0.30 \pm 0.13$ m a$^{-1}$, respectively.
Glacier mass balance can be acquired through glaciological, hydrological and geodetic
methods (Ye et al., 2015). Due to high altitude and harsh climatic conditions, it is hard to carry out
widespread in-situ measurements. Meanwhile, satellite remote sensing is a promising alternative
to conduct glacier mass balance through geodetic method, even in remote mountainous terrains
and over several glaciers at a time (Paul and Haeberli, 2008). Glaciers in the Kangri Karpo
Mountain were almost completely mapped by Topographic Maps that made from aerial
photographs in October 1980, and then mapped by X-band SAR Interferometry (InSAR) in
February 2000 during the SRTM resulting in a Digital Elevation Model. Glaciers in the region were
mapped again by single-pass X-band InSAR from TerraSAR-X on 18 February 2014 and 13 March
2014, and its add-on for digital elevation measurements (TanDEM-X) (Krieger et al., 2007). In
this study, the approach of Differential Synthetic Aperture Radar Interferometry (DInSAR) was
used to estimate glacier mass balance in the Kangri Karpo Mountain between 1980 and 2014.
**2   Study Area**
The Kangri Karpo Mountain, located on the eastern end of the Nyainqentanglha Mountains in
southeast Tibetan Plateau, extent about 280 km from northwest to southeast. This region located in
the south of Bomi County, and near to Motuo, Zayu and Basu County (Fig. 1). The north of this
region, Purlung Zangbo, is a tributary of the Yalung Zangbo River, and the other side is Gongri
Gabo River which belongs to west tributary of the Zayu River. Because the east section faces the
entrance of the moist southwest monsoon from Indian Ocean, which enters into the plateau at the
Grand Bend of the Yarlung Zangbo River, and the terrain forces the air flow to rise, it is the region
with the maximum precipitation and highest moisture on the plateau and hence glaciers are
well-developed (Shi et al., 2008a).



During winter and spring, the westerly jet in the Northern Hemisphere was blocked by
Tibetan Plateau and divided into two branches, southern branch develops into a trough in study
area after bypassing the Himalaya Mountains. Moisture in the Bay of Bengal region was attracted
by the trough, land on TP and form strong snowfall. In summer, due to the effect of topography
flattening, abundant precipitation was transported to study area by the Indian monsoon (Li et al.,
1986). Hence, the Kangri Karpo Mountain is the most humid region of Tibetan Plateau, and
become one of the most important and concentrated regions with maritime (temperate) glaciers
development (Shi and Liu, 2000). It is estimated that the mean summer air temperature at the
equilibrium-line altitude (ELA) of glaciers in the region is usually above 1 ℃, and annual
precipitation is about 2500–3000 mm (Shi et al., 1988). Most glaciers in study area are in the state
of pressure melting point, glacier surface ablation is intensive and the velocity of glacier
movement is fast (Li et al., 1986).

13       According to the first Chinese Glacier Inventory, the Kangri Karpo Mountain contains 1320
glaciers, with a total area and volume of 2655.2 $km^2$ and 260.3 $km^3$, respectively (Mi et al., 2002).
Yalong Glacier (CGI code: 5O282B37) is the largest one among these glaciers (191.4 $km^2$ in
surface area and 32.5 km long), while the Ata Glacier (CGI code: 5O291B181), located on the
south slope of the Kangri Karpo Mountain, is the glacier with the lowest terminus at 2450 m, 16.7
km in length and 13.75 $km^2$ in area (Liu et al., 2006). Comparison of photographs taken in
different time found that tongue position of Ata Glacier, ice volume and glacial surface conditions
have changed greatly in the past decades (Yang et al., 2008).
## 3   Data
### 3.1   Topographic Maps
Five topographic maps on 1:100000 scale and fifty topographic maps on 1:50000 scale generated
from aerial photos acquired in October 1980 by the Chinese Military Geodetic Service was
employed in the present study. Using a seven parameter transformation method, these maps
georeferenced into the 1954 Beijing Geodetic Coordinate System (BJ54) with geoid (datum level
is Yellow Sea mean sea level at Qingdao Tidal Observatory in 1956) were re-projected into World
Geodetic System 1984 (WGS1984)/Earth Gravity Model 1996 (EGM96) (Xu et al., 2013). The
contour lines of the maps were manually digitized and then converted into raster DEM (TOPO
DEM) with a 30 m grid cell by employing the Thiessen polygon method (Shangguan et al., 2010;
Wei et al., 2015b; Zhang et al., 2016a). According to the national photogrametrical standard of
China (GB/T12343.1, 2008), the vertical accuracy of the TOPO DEM is better than 9 m on
glaciers with gentle slopes (~19 °) which is common for most of the glacierized areas in the Kangri
Karpo Mountain.
### 3.2   Shuttle Radar Topography Mission
The SRTM was conducted to acquire interferometric synthetic aperture radar (InSAR) date
simultaneously in the C-band and X-band frequencies from 11 to 22 February 2000 (Farr et al.,
2007). The SRTM DEM can be referred to the glacier surface in the last balance year (1999) with
slight seasonal variances (Gardelle et al., 2013; Pieczonka et al., 2013; Zwally et al., 2011). The
X-band SAR system was operated with a swath width of 45 km leaving large data gaps in the
resulting X-band DEM (Rabus et al., 2003). Unfortunately, only 23% of the Kangri Karpo glaciers
are covered by the data set. The unfilled finished SRTM C-band DEM, with a swath width of 225



km and 1 arc-second resolution (approximately 30 m) in WGS84/EGM96, is freely available for
scientific purposes (http://earthexplorer.usgs.gov/). Hence, the unfilled finished SRTM C-band
DEM was employed to study ice surface elevation change.

### 3.3 TerraSAR-X/TanDEM-X

TerraSAR-X was launched in June 2007 by the German Aerospace Center (DLR). TerraSAR-X
and its add-on for digital elevation measurements (TanDEM-X) are flying in a close orbit
formation acting as a flexible single-pass SAR interferometer (Krieger et al., 2007).
Interferometric data acquisition can be performed in the pursuit monostatic mode, the bistatic
mode and the alternating bistatic mode. Current baseline for operational DEM generation is the
bistatic mode which minimizes temporal decorrelation and makes efficient use of the transmit
power (Krieger et al., 2007).
The experimental Co-registered Single look Slant range Complex (CoSSC) files, acquired in
bistatic InSAR stripmap mode on 18 February 2014 and 13 March 2014, were employed in this study
(Fig. 2 and Table 1). The CoSSC files have been focused and co-registered at the TanDEM-X
Processing and Archiving Facility (PAF). The GAMMA SAR and interferometric processing software
was employed for the interferometric processing of the CoSSC files (Werner et al., 2000).

### 3.4 Landsat images

In order to analyze the relationship between glacier mass balance and changes in glacier extent, two
Landsat Thematic Mapper (TM) scenes, one Landsat Enhanced Thematic Mapper Plus (ETM+) scene
and three Landsat Operational Land Imager (OLI) scenes were employed in this study (Table 1). It is
better that Landsat images were acquired in the same year with SRTM and TerraSAR-X/TanDEM-X
acquisitions, while because of the low quality of Landsat images in 2000 and 2014, Landsat
TM/ETM+ images in 2001 and Landsat OLI images in 2015 with high quality were selected in
this study. All Landsat images are available from the United States Geological Survey (USGS), and are
orthorectified with the SRTM and ground control points from the Global Land Survey 2005 (GLS2005)
dataset. Amongst the Landsat images, the co-registered TerraSAR-X coherence image, the SRTM-X
DEM and Topographic Maps, almost no horizontal shift was observed. For Landsat ETM+/OLI images,
pan-sharpening employing principal component analysis was performed to enhance the spatial
resolution to 15 m.

### 4   Methods

#### 4.1 Glacier Inventory

The outlines of glaciers in October 1980 were delineated manually from Topographic Maps. These
maps were geo-referenced and rectified with a kilometer grid, and validated by reference to the
original aerial photographs to update the first Chinese Glacier Inventory (Wu et al., 2016b).
Global inventory of glacier outlines can be available through the Randolph Glacier Inventory
(Arendt et al., 2015). For the Kangri Karpo Mountain, these glacier outlines are taken from "the
Second Chinese Glacier Inventory" (CGI2) that delineated from Landsat TM on 8 September
2005 (Guo et al., 2015). An inventory of high-mountain Asian glaciers named "Glacier Area
Mapping for Discharge from the Asian Mountains" was compiled from 356 Landsat ETM+ scenes
in 226 path-row sets (Nuimura et al., 2015). The GAMDAM outlines are nearly all from
1999-2003 and thus conform with the recommendation for the compilation of glacier inventory to

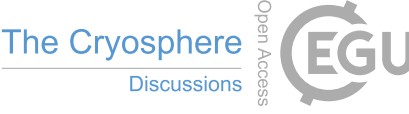



select imagery as close to 2000 as possible (Arendt et al., 2015; Paul et al., 2009). Hence, Landsat TM/ETM+ scenes were used to validate and update the CGI2 and GAMDAM glacier inventory, and generated the 2000 inventory of the detailed study area.

A semi-automated approach using the TM3/TM5 band ratio was applied to delineate glacier outlines in 2015 using Landsat OLI images (Bolch et al., 2010b; Paul et al., 2009; Racoviteanu et al., 2009). To ensure that ice patches were larger than 0.01 km2, a 3 by 3 median filter was applied to eliminate isolated pixel (Bolch et al., 2010b; Wu et al., 2016b). Then, the derived glacier polygons are checked manually against images from adjacent years with less or no snow and cloud-free, to discriminate proglacial lakes, seasonal snow, boulders on the glacier and debris-covered ice (Fig. 3). The final contiguous ice coverage was divided into individual glacier polygons using topographical ridgelines (TRLs), which were automatically generated based on the SRTM-C DEM (Guo et al., 2011).

The best way to assess the accuracy of glacier outlines is to compare extracted results with independently digitized glacier outlines from high resolution aerial imagery at random locations (Bolch et al., 2010a; Paul et al., 2003). Previous studies indicated that the average offsets between glacier outlines derived from Topographic Maps and Corona-image outlines was ±6.8 m (Wu et al., 2016b), and the average offsets between Landsat-image outlines and real-time kinematic differential GPS (RTK-DGPS) measurements, Google Earth$^{TM}$ images with a spatial resolution better than 1 m, were ±10 m and ±30 m for the delineation of clean and debris-covered ice, respectively (Guo et al., 2015). On the basis of these average offsets, mean relative errors of ±1.3%, ±2.0% and ±2.4% were calculated for glacier area in 1980, 2000 and 2015, respectively.

## 4.2 Glacier Length

Glacier length is a key parameter in glacier inventory and its vector representation (glacier centerlines) is important to model future glacier evolution and calculate glacier ice volume (Le Bris and Paul, 2013). Some researchers defined it as the central flowline from the highest glacier elevation to the terminus, whereas others regard the length of the longest flowline as glacier length (Kienholz et al., 2014; Leclercq et al., 2014). The length change can be calculated by intersecting the central flowline with the respective glacier outline (Paul and Svoboda, 2010), or calculated as the average length from the intersection of the glacier outlines with the stripes which drawn parallel to the main flow direction of the glaciers (Koblet et al., 2010).

In this study, a new strategy based on a glacier axis concept from glacier morphology perspective was applied that requires only glacier outlines and a DEM as input (Yao et al., 2015). An error estimation of the resulting glacier centerlines was performed that compared the semi-automatically generated results to high resolution aerial imagery at the terminus of glacier centerlines. A Corona image with a resolution of 4 m and Google Earth$^{TM}$ images with a resolution better than 1 m were used to evaluate the accuracy of glacier centerlines. Hence, the uncertainties of glacier centerlines from Topographic Maps and Landsat images are no more than 6 m and 7.5 m, respectively.

## 4.3 Glacier elevation changes

The TerraSAR-X/TanDEM-X acquisitions were processed by differential SAR interferometry (DInSAR) (Neckel et al., 2013) using GAMMA SAR and interferometric processing software (Werner et al., 2000). In order to improve the phase-unwrapping procedure and minimize errors, the unfilled finished SRTM C-band DEM were employed in this study. Before generating the





differential interferogram, precise horizontal offset registration and fitting between the SRTM C-band DEM and the TerraSAR-X/TanDEM-X acquisitions is necessary. Based on the relation between the map coordinates of the SRTM C-band DEM segment covering the TerraSAR-X/TanDEM-X master file, and the SAR geometry of the respective master file, an initial lookup table was calculated. While the areas of radar shadows and layover in the TerraSAR-X/TanDEM-X interferogram would induce gaps in the lookup table, a method of linear interpolation between the gap edges in each line of the lookup table was used to fill these gaps. The offsets between the master scene and the simulated intensity of the SRTM C-band DEM, were calculated using cross correlation optimization of the simulated SAR images employing *GAMMA's offset_pwrm* module. The horizontal registration and geocoding lookup table were refined by these offsets. The SRTM C-band DEM was translated from geographic coordinates into SAR coordinates via the refined geocoding lookup table, and conversely, the final difference map was translated from SAR coordinates into geographic coordinates. Then a differential interferogram was generated by the TerraSAR-X/TanDEM-X interferogram and the simulated phase of the co-registered SRTM C-band DEM. An adaptive filtering approach was used to filter the differential interferogram (Goldstein and Werner, 1998), and then *GAMMA's* minimum cost flow (MCF) algorithm was employed to unwrap the flattened differential interferogram. According to the computed phase-to-height sensitivity and select ground control points (GCPs) from the respective off-glacier pixel locations in the SRTM C-band DEM, the unwrapped differential phase was converted to absolute differential heights. While, a residual not covered by the baseline refinement would be existed, and can be regarded as a linear trend that estimated by a two dimensional first order polynomial fit in off-glacier regions. The linear trend and a constant vertical offset were removed from the maps of absolute differential heights. Finally, the resulting data sets were translated to a metric cartographic coordinate system with 30 m $\times$ 30 m pixel spacing (Neckel et al., 2013). The same method of DInSAR was employed to acquire the glacier elevation change from 1980 to 2014 with the data sets of TOPO DEM and TerraSAR-X/TanDEM-X acquisitions.

For the glacier elevation change from 1980 to 1999, a difference map was constructed by common DEM differencing with TOPO DEM and SRTM C-band DEM (Bolch et al., 2011; Nuth and Kääb, 2011; Pieczonka et al., 2013; Wei et al., 2015a). Relative horizontal and vertical distortions between the two data sets, can be corrected by statistical approaches that based on the relationship between elevation difference and slope, aspect (Nuth and Kääb, 2011). Elevation differences in off-glacier regions were used to analyze the consistency of TOPO DEM and SRTM C-band DEM (Fig. 4). Outliers of elevation differences, usually around data gaps and near DEM edges, were omitted with values exceeding $\pm 100$ m (Berthier et al., 2010; Bolch et al., 2011). The vertical biases and horizontal displacements could be adjusted simultaneously using the substantial cosinusoidal relationship between standardized vertical bias and topographical parameters (slope and aspect). And the biases that caused by different spatial resolutions between DEMs, could be adjusted by the relationship between elevation differences and maximum curvatures (Gardelle et al., 2012a; Nuth and Kääb, 2011).

The penetration depth of SRTM C-band radar beam into snow and ice need to be considered for elevation changes of glacier surfaces (Berthier et al., 2006; Gardelle et al., 2012a; Pieczonka et al., 2013). Penetration depth can range from 0 to 10 m depending on a variety of parameters such as snow temperature, density and water content (Berthier et al., 2006; Dall et al., 2001). As a first





approximation, the penetration depth of SRTM X-band radar beam is much smaller than C-band,
the elevation difference between these two data sets can be considered as the penetration of SRTM
C-band radar beam into snow and ice (Gardelle et al., 2012a). The elevation differences between
SRTM C-band and X-band indicate that the penetration depth of the C-band averaged 1.24 m
in the study area. The mean value of the SRTM C-band penetration over glaciers in Kangri
Karpo Mountain is in agreement with (Gardelle et al., 2013), who found penetration of 1.7 m
in eastern Nyainqentanglha Mountains (named Hengduan Shan in(Gardelle et al., 2013)).
**4.4  Mass balance and error estimation**
In order to convert the derived surface elevation changes into the mass balance of glaciers, a
density of ice/firn/snow should be considered. A value of 900 kg m$^{-3}$ was applied to assess the
mass changes in water equivalent (w.e.) from elevation differences, and then adding an ice density
uncertainty of 17 kg m$^{-3}$.
For the accuracy assessment of TOPO DEM and SRTM C-band DEM, the elevation product
of Geoscience Laser Altimeter System (GLAS) carried on-board the Ice Cloud and Elevation
Satellite (ICESat) was utilized in this study (Neckel et al., 2013). All available GLAS elevation
data for the study area were obtained from the National Snow and Ice Data Center (NSIDC)
(release 634; product GLA 14). Because of the effect of clouds during the time of data acquisition,
some GLAS elevation data cannot represent the true altitude of ground surface. Outliers of
elevation difference between GLA 14 and multi-source of DEMs in off-glacier regions were
removed from the analysis with values exceeding ±100 m. Compared to the GLAS elevation data,
a mean and standard deviation of 2.74 ± 1.73 m and 2.65 ± 1.48 m for TOPO DEM and SRTM
C-band   DEM,   respectively.   Due   to   the   GCPs   that   convert   the   unwrapped
TerraSAR-X/TanDEM-X interferogram into absolute heights was selected from the respective
off-glacier pixel locations in the SRTM C-band DEM, vertical bias of TerraSAR-X/TanDEM-X
DEM and GLA 14 is similar with the bias of SRTM C-band DEM and GLA 14.
For an error estimate of the derived surface elevation changes, the residual elevation
differences were estimated in off-glacier regions assuming that these areas did not change in
height between 1980 and 2014 and that elevations should be equal in TOPO DEM, SRTM C-band
DEM and TerraSAR-X/TanDEM-X DEM. The mean elevation difference (MED) over off-glacier
regions between the final difference maps was in the range of -1.42 to 0.75 m (Table 2). Because
averaging over larger regions reduces the errors, the standard deviation (STDV) over off-glacier
regions would probably overestimate the uncertainty for larger samples. Thus, the uncertainty can
be estimated by the standard error of the mean (SE):

$$SE = \frac{STDV}{\sqrt{N}} \tag{1}$$

where $N$ is the number of the included pixels. To avoid the effect of autocorrelation,
de-correlation length of 600 m and 200 m was employed for difference maps that derived by
common DEM differencing and DInSAR (Bolch et al., 2011; Neckel et al., 2013). Then, the
overall errors of the derived surface elevation changes can be estimated using SE and MED
over off-glacier regions:

$$\sigma = \sqrt{MED^2 + SE^2} \tag{2}$$

Finally, the root of sum of squares of the estimated errors of glacier area and surface
elevation changes, and ice density uncertainty of 17 kg m$^{-3}$, were used to estimate the overall
errors of mass balance (Neckel et al., 2013).





## 5  Results

### 5.1  Area change

According to the 2015 inventory, the Kangri Karpo Mountain contains 1166 glaciers, with an area of 2048.50 ± 48.65 km$^2$, and the mean glacier size is about 1.76 ± 0.04 km$^2$ (Fig. 5). The highest number of glaciers can be found in the size class 0.1-0.5 km$^2$, whereas glaciers between 1-5 km$^2$ cover the largest area (Fig. 5A). Only two glaciers area larger than 50 km$^2$, the largest glacier is Yalong Glacier and another is Xirinongpu Glacier, with glacier area of 173.00 ± 0.67 km$^2$ and 90.28 ± 0.23 km$^2$, respectively. Glaciers area in Kangri Karpo Mountain present a normal distribution in different elevation, about 76.9% of glacier area lies in the 4500-5500 m elevation range. Azha Glacier is the glacier with the lowest glacier tongue position, and the elevation of glacier tongue at 2551 m (Fig. 5B). Median elevation of the glaciers in Kangri Karpo Mountain is situated at around 4852 m, 5215 m for glaciers on the north slope and 4639 m on the south. This is consistent with the tendency of equilibrium line altitude in southeast Tibetan Plateau (Su et al., 2014). The mean glacier surface slope in the Kangri Karpo Mountain is 24.1 °, with most in the 12 °-32 ° range that accounts for 80.5% of the glaciers and 85.4% of their area. Most glaciers have a SE, S or SW aspects, account for 59.2% and 80.9% of glacier number and area, respectively.

Comparing the total area of all glaciers in 1980 with that in 2015, ice cover in the Kangri Karpo Mountain diminished by 679.51 ± 59.49 km$^2$ (24.9% ± 2.2%) or 0.71% ± 0.06% a$^{-1}$. Small glaciers shrunk in area in larger percentage (Fig. 5C). Meanwhile, absolute area loss was higher for larger glaciers. Analysis of the glacier hypsography indicated that ice coverage disappeared completely below 2500 m, the highest absolute area loss occurred in the 4500 – 4700 m a.s.l. altitude range, and ice coverage remains almost unchanged above 5800 m. The average minimum elevation of the glaciers increased by 106 m, while median elevation rose about 56 m from 4796 to 4852 m.

There was a slight tendency that the rate of glaciers shrinkage from 1980 – 2000 was lower than those from 2000 – 2015 in the detailed study area of Kangri Karpo Mountain (Table 3). In the period of 1980 – 2000, glacier area decreased by 63.72 ± 9.06 km$^2$ from its original 784.60 km$^2$ (8.1% ±1.2%) or 0.41% ± 0.06% a$^{-1}$. Between 2000 and 2015, glaciers experienced a reduction of 56.00 ±10.97 km$^2$ (7.8% ±1.5%) or 0.52% ±0.10% a$^{-1}$. A detailed analysis of ten sample glaciers confirmed that all glaciers decreased continuously throughout all investigated periods (Table 4). Percentage area loss varied between 8.6% (WGI ID/GLIMS ID: 5O291B0200/G097005E29155N, the smallest glacier loss) and 20.9% (Parlung No. 10 Glacier, the largest one) from 1980 – 2015. While the largest loss of absolute area reached 20.43 km$^2$ for Yalong Glacier, and lowest reached 1.04 km$^2$ for Parlung No. 10 Glacier.

### 5.2  Length change

Comparing the terminal of all glaciers, only nine glaciers advanced while others retreated in the Kangri Karpo Mountain from 1980 – 2015. The nine glaciers experienced a mean advance of 14.8 m a$^{-1}$, and the length of centerlines increased between 103 m and 1547 m. The terminus elevation decrease of these advanced glaciers averaged at 191 m, varying between 34 m (the smallest lowering from 4796 m to 4762 m a.s.l. altitude) and 412 m (the largest one from 4362 m to 3949 m a.s.l. altitude) (Table 5). Based on different glacier size, slope and aspect, 86 glaciers were selected from all retreated glaciers to analyze the changes of length. These selected glaciers experienced a mean recession of 759 m (21.7 m a$^{-1}$) with smallest of 6 m and largest of 3956 m.




Similar to area change of the glaciers in the case study region, these glaciers have shown accelerated retreat during the two periods from 1980 – 2000 and from 2000 – 2015 as measured in glacier length (Table 6). Glaciers experienced a mean length reduction of 21.0 m a$^{-1}$ (varying from 2.5 m a$^{-1}$ to 104.2 m a$^{-1}$) in the period of 1980 – 2000 and 22.6 m a$^{-1}$ from 2000 – 2015 (varied between 1.3 m a$^{-1}$ and 144.8 m a$^{-1}$). The recession of Yalong Glacier changed from 78.0 m a$^{-1}$ in 1980 – 2000 to 13.6 m a$^{-1}$ in 2000-2014, while the average rates of retreat increased significantly for Azha Glacier in the two sub-periods of 1980 – 2000 and 2000 – 2015, with a mean recession of 11.3 m a$^{-1}$ and 144.8 m a$^{-1}$, respectively.

### 5.3 Mass balance

The average elevation change of the entire glacier surfaces in the case study area of Kangri Karpo Mountain was -17.46 $\pm$ 0.54 m from 1980-2014. The glaciers, with an area of 788.28 km$^2$, have experienced a mean thinning of 0.51 $\pm$ 0.09 m a$^{-1}$, or mean mass deficit of 0.46 $\pm$ 0.08 m w.e. a$^{-1}$, equivalent to an overall mass change of -13.76 $\pm$ 0.43 Gt for the case study area during 1980-2014. Thinning of these glaciers has speeded up during the periods of 1980-2000 and 2000-2014. From 1980-2000, glaciers have thinned by 5.30 $\pm$ 0.77 m on average, and experienced a mass loss of 0.24 $\pm$ 0.16 m w.e. a$^{-1}$. Then a surface lowering of 11.04 $\pm$ 0.43 m was found from 2000-2014, with a mass loss of 0.71 $\pm$ 0.10 m w.e. a$^{-1}$ (Fig. 6 and Table 7).

Heterogeneous glacier mass balance was presented in the detailed study area of Kangri Karpo Mountain during 1980-2014. Glaciers, with an area of 471.05 $\pm$ 3.03 km$^2$ in the drainage basin of 5O282B, experienced greater mass deficit of 0.51 $\pm$ 0.22 m w.e. a$^{-1}$ from 1980-2014, and mean mass loss of 0.30 $\pm$ 0.14 m w.e. a$^{-1}$ and 0.76 $\pm$ 0.22 m w.e. a$^{-1}$ during the periods of 1980-2000 and 2000-2014, respectively. Mean mass deficit of 0.39 $\pm$ 0.11 m w.e. a$^{-1}$ in the drainage basin of 5O291B was smaller than those in 5O282B drainage basin during 1980-2014. Glaciers with an area of 317.22 $\pm$ 4.27 km$^2$ in 5O291B drainage basin experienced accelerating mass loss during the periods of 1980-2000 and 2000-2014, with mean mass loss of 0.13 $\pm$ 0.16 m w.e. a$^{-1}$ and 0.63 $\pm$ 0.04 m w.e. a$^{-1}$.

Ablation area and accumulation area both experienced mass loss in the detailed study area of Kangri Karpo Mountain from 1980-2014, with mean mass loss of 0.73 $\pm$ 0.08 m w.e. a$^{-1}$ and 0.32 $\pm$ 0.08 m w.e. a$^{-1}$, respectively. Glaciers in ablation area experienced accelerating mass loss from 1980-2014, while the rate of mass loss remain stable in accumulation area during the whole period. The latter period since 2000 has seen a mass loss of 1.35 $\pm$ 0.04 m w.e. a$^{-1}$, almost 4 folds (0.27 $\pm$ 0.16 m w.e. a$^{-1}$) before 2000. Mass losses in accumulation area were similar during 1980-2000 and 2000-1014, with values of 0.22 $\pm$ 0.14 m w.e. a$^{-1}$ and 0.37 $\pm$ 0.22 m w.e. a$^{-1}$.

Prominent thickening (elevation increase) was found on the termini of two glaciers on the southern slope of the Kangri Karpo Mountain (Fig. 6C, WGI: 5O291B0113 and 5O291B0117). Probably the effect of debris cover, the glacier terminus of 5O291B0113 and 5O291B0117 remains stable between October 1980 and October 2015.

## 6 Discussion

### 6.1 Uncertainty

The uncertainty of glacier outlines was caused by positional and processing errors associated with glacier delineation (Bolch et al., 2010a; Racoviteanu et al., 2009). Seasonal snow, cloud and debris cover complicated glacier mapping precisely (Paul et al., 2013). The accuracy of glacier outlines in this study was estimated by compare extracted results with independently digitized



glacier outlines from high resolution aerial imagery at random locations. An uncertainty model was employed to assess accuracy estimated in this study (Pfeffer et al., 2014). The delineation uncertainty of glaciers in the Kangri Karpo Mountain in 2015 was about 24.33 km$^2$ using the uncertainty model, that is smaller than the uncertainty of 48.65 km$^2$ in this study. Main reason for this discrepancy is probably that the delineation uncertainty of glaciers have been overestimated in this study, especially in the area of debris-covered ice and exposed bedrock that surrounded by ice cover.

For the uncertainty of mass balance, the penetration depth of SRTM C-band radar beam into snow and ice was critical issue when SRTM DEM was employed for geodetic mass balance calculations. The penetration depth of SRTM C-band radar beam can be estimated by comparing the SRTM C-band with the SRTM X-band DEM (Gardelle et al., 2012a; Kääb et al., 2012). Previous studies indicated that the penetration depth decreases as temperature and water content of surface snow cover rise (Surdyk, 2002), and penetration depths at 10 GHz from 2.1 m to 4.7 m were measured in Antarctica (Davis and Poznyak, 1993). Glaciers in eastern Nyainqentanglha Mountain are predominantly monsoonal influenced and have more snow moisture and higher temperatures than the Antarctic ice sheet (Shi and Liu, 2000). Hence, the penetration correction is suitable under the assumption that the influence of slight penetration of the X-band is negligible. The mean SRTM C-band penetration was 1.24 m in Kangri Karpo Mountain, led to mass changes on average of +0.06 m w.e. a-1 and –0.08 m w.e. a-1 for the periods of 1980-2000 and 2000-2014.

Another issue is the lack of information in several regions due to data voids in accumulation area. Different suitable assumptions or elevation changes in the accumulation regions were used to fill the data voids and to assess the impact on mass balance (Pieczonka et al., 2013; Shangguan et al., 2015). In this study, the information of elevation change exists in all altitudinal zones from 2400 m to 6600 m a.s.l., and the area of data voids was too small (0.7% above 6000 m a.s.l. in area) to affect the mass balance significantly. Hence, data voids can be neglected when mass balance was estimated by all glaciers area in the detailed study area, average surface elevation change and ice density.

### 6.2  Glacier change of area and length

This study found glacier shrinkage in Kangri Karpo Mountain between 1980 and 2014 of about 0.71% ±0.06% a$^{-1}$. In the period of 1980 – 2000, glacier area decreased by 0.41% ±0.06% a$^{-1}$ in the case study area, and then this value increased to 0.52% ±0.10% a$^{-1}$ after 2000. Our result is in agreement with previous study, that found shrinkage of 0.57% a-1 in the southeastern Tibetan Plateau from 1980-2001 (Yao et al., 2012a). The shrinkage rate of previous study is larger than our result slightly, main reason for that is probably the difference of glacier size. The mean glacier sizes are about 2.07 km$^2$ and 6.54 km2 in the Kangri Karpo Mountain and in the detailed study area in 1980, and greater relative loss for smaller glaciers was found in this study and previous studies (Wei et al., 2014; Wu et al., 2016b).

Compared with the retreat of mountain glaciers in the western China, glaciers in Kangri Karpo Mountain have experienced extremely strong glacial retreat. The glacier retreat of about 0.71% a$^{-1}$ is lower than that in Altay Mountain (0.75% a$^{-1}$) (Yao et al., 2012b), but larger than that in other regions of western China, such as Tian Shan Mountains (0.22% a$^{-1}$) (Wang et al., 2011), eastern Pamir (0.25% a$^{-1}$) (Zhang et al., 2016b), wester Kunlun Mountain (0.09% a$^{-1}$) (Bao et al., 2015), Qilian Mountain (0.47% a$^{-1}$) (Sun et al., 2015), Tibetan Plateau interior area (0.26% a$^{-1}$) (Wei et al., 2014).





The location of glacier terminal is often measured by remote sensing and investigation in the
field. Due to the differences in the periods studied and spatial scales, the length changes of glacier
centerlines in this study are slower than previous studies, except for Azha Glacier (Liu et al., 2006;
Yang et al., 2010; Yao et al., 2012a). The investigation of terminus variation on Parlung No. 10
Glacier was surveyed from 2006-2008, the period of survey is too short to represent length change
of glacier centerlines over long periods of time (Yang et al., 2010). Compared the length change of
Azha Glacier in different periods, -56.1 m a$^{-1}$ from 1973-2005 (Yao et al., 2012a), -65 m a$^{-1}$ from
1980-2006 (Yang et al., 2010) and -70 m a$^{-1}$ from 1980-2015 (this study), we can found that Azha
Glacier have experienced greater retreat after 2000s than that before 2000s. The length change of
Yalong Glacier from 1980-2000 in this study is similar with Liu et al. (Liu et al., 2006), who
found retreat of 73 m a$^{-1}$ of Yalong Glacier between 1980 and 2001. And then the average rate of
retreat decreased significantly for Yalong Glacier after 2000. Main reason is probably that flow
velocity increased and more ice and snow are transported from accumulation area to glacier
terminal.
For advanced glaciers, the mean glacier size is about 0.51 km$^2$ and mean glacier surface slope
is about 27.9 °, most glaciers have an S or SW aspect, and mean accumulation area ratio (AAR) is
51. Previous studies also found advanced glaciers in Kangri Karpo Mountain (Liu et al., 2006; Shi
et al., 2006). Compared with the CGI2 and GAMDAM glacier inventory, the location of most
glacier terminals in 2000 and 2014 are closed, indicated that glacier advancing mainly occurs
before 2000. This behavior may be related to the increase of high precipitation (Shi et al., 2006),
or a phenomenon of glacier surging, and requires further investigation.

## 6.3  Glacier thinning and mass balance

A comparison of glacier thickness changes show that significant differences are found over
eastern Nyainqentanglha Mountain. Using SRTM DEM and SPOT5 DEM (24 November 2011),
glaciers experienced a mean thinning of 0.39 ±0.16 m a$^{-1}$ in eastern Nyainqentanglha Mountains
(named Hengduan Shan) (Gardelle et al., 2013). Based on ICESat and SRTM, (Kääb et al., 2015),
(Neckel et al., 2014) and (Gardner et al., 2013) acquired different results over eastern
Nyainqentanglha Mountain, with glacier thickness loss of 1.34 ±0.29 m a$^{-1}$, 0.81 ±0.32 m a$^{-1}$ and
0.30 ±0.13 m a$^{-1}$, respectively. Using SRTM DEM and TerraSAR-X/TanDEM-X acquisitions (18
February 2014 and 13 March 2014), glaciers experienced a mean thinning of 0.79 ±0.11 m a$^{-1}$ in
Kangri Karpo Mountain. At a first glance, the result of glacier thickness loss in this study is in
agreement with (Neckel et al., 2014), and have significant differences with (Kääb et al., 2015).
Main reason for this discrepancy is the different estimation of SRTM C-band penetration. An
average SRTM C-band penetration of 1.24 m was used for Kangri Karpo Mountain that estimated
from the difference of SRTM C-band and X-band DEMs (Gardelle et al., 2012a). While an
average penetration of 8-10 m (7-9 m if based on the winter trends that might alternatively be
assumed to reflect February conditions) was employed for eastern Nyainqentanglha Mountain
(Kääb et al., 2015). Previous studies indicated that the penetration depth varies with temperature
and water content (Surdyk, 2002), and penetration depths of SRTM C-band from 1.4 m to 3.4 m
were estimated over the Pamir-Karakoram-Himalaya (Gardelle et al., 2013; Kääb et al., 2012).
The characteristics of glaciers in eastern Nyainqentanglha Mountain are similar with that in
eastern Himalaya (Shi et al., 2008b). Therefore, the penetration in this study is more suitable
under the assumption that the penetrations in eastern Nyainqentanglha Mountain and eastern
Himalaya are similar.





Field measurement of mass balance is the best indicator of glacier change. A monitoring
program has been carried out on Parlung No. 4 Glacier (5O282B0004/G096920E29228N) and
Parlung No. 10 Glacier (5O282B0010/G096904E29286N), both located on the northern slope of
the Kangri Karpo Mountain. Large ice deficit were found on the two monitored glaciers at rates of
-0.71 m w.e. a$^{-1}$ from May 2006 to May 2007 and -0.78 m w.e. a$^{-1}$ during 2005-2009, respectively
(Yang et al., 2008; Yao et al., 2012a). Based on SRTM DEM and TerraSAR-X/TanDEM-X
acquisitions (18 February 2014), the two glaciers experienced large surface lowering from 2000 to
2014, with mean mass loss of 0.65 $\pm$ 0.22 m w.e. a$^{-1}$ and 0.67 $\pm$ 0.22 m w.e. a$^{-1}$. The comparison
between filed measurements of mass balance and the result of this study indicated that there was
high consistency of glacier mass loss about Parlung No. 4 Glacier and Parlung No. 10 Glacier.
Interestingly is the bigger thinning on the debris-covered regions of -0.99 $\pm$ 0.09 m a$^{-1}$ on
average than clean-ice region of -0.89 $\pm$ 0.08 m w.e. a$^{-1}$ from 1980-2014 (Fig. 7). Due to complex
surface conditions, such as supraglacial lakes, ice cliffs and heterogeneity of debris cover, the
mass loss patterns on the debris-covered tongue are complicated (Pellicciotti et al., 2015). It is
generally believed that ice ablation rate is highly reduced with the thick debris-cover due to the
insulation effect of debris. However, previous studies found that glacier ablation on debris-covered
regions were greater than on the exposed ice regions (Pu et al., 2003; Ye et al., 2015; Zhang et al.,
2016a). The location of debris-covered regions, in the lower altitudes with higher temperature, and
the development of supraglacial lakes and ice cliffs are probably the reasons for the larger mass
loss on the debris-covered regions (Benn et al., 2012; Sakai and Fujita, 2010).
Overall, negative elevation changes were found in glacier tongue regions except for two
glaciers in the southern slope of the Kangri Karpo Mountain (Fig. 6C). Compared the average
elevation changes of the two glaciers tongue surface during 1980-2000 and 2000-2014, positive
elevation changes were found between October 1980 and February 2000, and negative elevation
changes were found after 2000. The two glaciers showed positive elevation changes at the
terminuses, while elevation changes are unknown in accumulation area of the two glaciers due to
data voids. This phenomenon could be interpreted as glacier surging (Cuffey and Paterson, 2010),
or result from the increase of high precipitation (Shi et al., 2006).
**6.4 Climatic considerations**
The climate in Kangri Karpo Mountain is characterized be the westerly in winter and the
Indian monsoon in summer (Li et al., 1986). While the effects of westerly are weak in the study
area due to the block of Tibetan Plateau. Hence, the accumulation on glaciers in Kangri Karpo
Mountain is supplied by summer monsoon precipitation (Bolch et al., 2010a; Yao et al., 2012a).
Previous studies indicated that the Tibetan Plateau has experienced an overall surface air warming
since the mid-1950s (Duan et al., 2015; Li et al., 2010; Liu et al., 2008; Liu et al., 2009; Qin et al.,
2009; Yang et al., 2014; Yao et al., 2012a; You et al., 2010). Different trend of average annual
temperature results from different data in the southeastern TP. Based on temperature data from
meteorological stations, the southeastern TP present the lowest warming rate (Duan et al., 2015),
while the most warming rate was found from the MODIS land surface temperature (MODIS LST)
in the southeastern TP (Yang et al., 2014), even decreasing trend of average annual temperature
was found from the National Centers for Environmental Prediction/National Center for
Atmospheric Research (NCEP/NCAR) Reanalysis data (You et al., 2010). The changes in air
temperature were accompanied by the changes in precipitation due to variations in monsoonal
activity. Based on the Global Precipitation Climatology Project (GPCP) data, precipitation

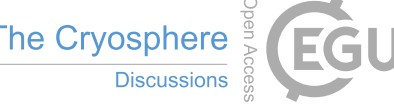

decreased in the southeastern TP from 1979 to 2010 (Yao et al., 2012a). While positive trend of annual precipitation was found from Chinese meteorological stations data in the southeastern TP, precipitation amount increased and the frequency of severely dry events decreased significantly (Li et al., 2010). Due to the ambiguous of climate change, the presented glacier changes and mass balance cannot be explained by the above summarized climate variations directly.

In order to analyze the response of glaciers to climate change, air temperature and precipitation datasets were collected from the China Meteorological Forcing Dataset (CMFD, 1979.01.01 – 2012.12.31) (Chen et al., 2011; He and Yang, 2011), which produced by merging a variety of data sources, including meteorological station data, TRMM satellite precipitation analysis data, GEWEX-SRB downward shortwave radiation data and GLDAS data (http://westdc.westgis.ac.cn/data/7a35329c-c53f-4267-aa07-e0037d913a21). The horizontal distributions of surface temperature change and precipitation change from May to September derived from the CMFD data was shown in Fig. 8. It is clear that warming is a dominant phenomenon in the southeastern TP during recent decades. The warming rate on the northern slope of the Kangri Karpo Mountain is larger than that on the southern slope slightly. The law of precipitation change was inconsistent that an increasing trend was present in much of Kangri Karpo Mountain, but a decreasing trend in the eastern Kangri Karpo Mountain. The changes of air temperature and precipitation are confirmed by three nearest meteorological stations datasets, which is Bomi (2736 m a.s.l.), Zuogong (3780 m a.s.l.) and Zayu (2423 m a.s.l.) (Liu et al., 2006; Yang et al., 2010). The air temperature at the three meteorological stations increased slightly from 1980-2000, and then it increased significantly after 2000. Despite large inter-annual fluctuation of precipitation, statistically significant trends are not evident at the three stations (Wu et al., 2016a; Yang et al., 2010). Hence, glaciers change in the Kangri Karpo Mountain can be attributed to climate warming, especially the rate of glaciers shrinkage and mass loss from 1980 – 2000 were lower than those from 2000 – 2015, and mean mass deficit in northern slope was larger than those in southern slope during 1980-2014, which is consistent with the tendency of climate warming.

## 7 Conslusions

This study estimated glacier area, glacier length, surface elevation and mass balance of the Kangri Karpo Mountain for the period of 1980-2015 based on Topographic Maps, Landsat images, SRTM and TerraSAR-X/TanDEM-X acquisitions.

Our results show that the Kangri Karpo Mountain contains 1166 glaciers, with an area of 2048.50 $\pm$ 48.65 km$^2$ in 2015. Ice cover in the Kangri Karpo Mountain diminished by 679.51 $\pm$ 59.49 km$^2$ (24.9% $\pm$ 2.2%) or 0.71% $\pm$ 0.06% a$^{-1}$ from 1980-2015. Comparing the terminal of all glaciers, only nine glaciers advanced while others retreated in the Kangri Karpo Mountain from 1980 – 2015. Compared with the retreat of mountain glaciers in the western China, glaciers in Kangri Karpo Mountain have experienced extremely strong glacial retreat.

The average elevation change of the entire glacier surfaces in the detailed study area of Kangri Karpo Mountain was -0.51 $\pm$ 0.09 m a$^{-1}$, or mean mass deficit of 0.46 $\pm$ 0.08 m w.e. a$^{-1}$ from 1980-2014. Heterogeneous glacier mass balance was presented in the Kangri Karpo Mountain during 1980-2014. The comparison between filed measurements of mass balance and the result of this study indicated that there was high consistency of glacier mass loss about Parlung No. 4 Glacier and Parlung No. 10 Glacier. Geodetic mass balance measurements in the detailed study area of Kangri Karpo Mountain revealed that the debris-covered regions exhibit higher thinning rates than the clean-ice region on average obviously, with an average of -0.99 $\pm$ 0.09 m





a$^{-1}$ (-0.89 $\pm$ 0.08 m w.e. a$^{-1}$) from 1980-2014. There was a slight tendency that the rate of glaciers shrinkage and mass loss from 1980 – 2000 were lower than those from 2000 – 2015 in the detailed study area of Kangri Karpo Mountain.

*Acknowledgements.* This work was supported by the fundamental program from the Ministry of Science and Technology of China (MOST) (Grant No. 2013FY111400), the National Natural Science Foundation of China (Grant No. 41190084) and the International Partnership Program of Chinese Academy of Sciences (Grant No. 131C11KYSB20160061). Landsat images are from the U. S. Geological Survey and NASA. The GAMDAM glacier inventory is from Dr. A. Sakai . The first and second glacier inventories were provided by an immediate past MOST project (2006FY110200). The China Meteorological Forcing Dataset (CMFD) is from Cold and Arid Regions Science Data Center at Lanzhou. All SAR processing was done with the GAMMA SAR and interferometric processing software.

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



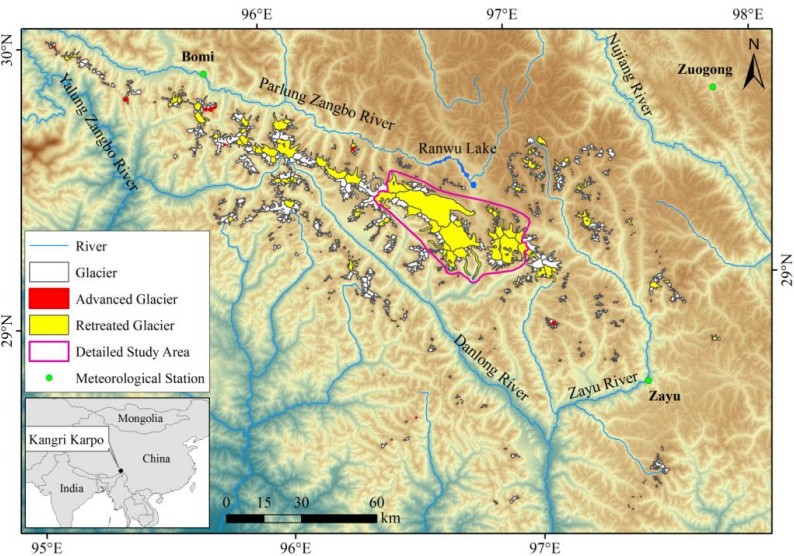

Figure 1. Overview of study area and glacier distribution, including the locations of the detailed
study area and meteorological stations. 96 glaciers were selected to generate centerlines and
calculate length change, and then be distinguished into advanced glaciers and retreated glaciers.

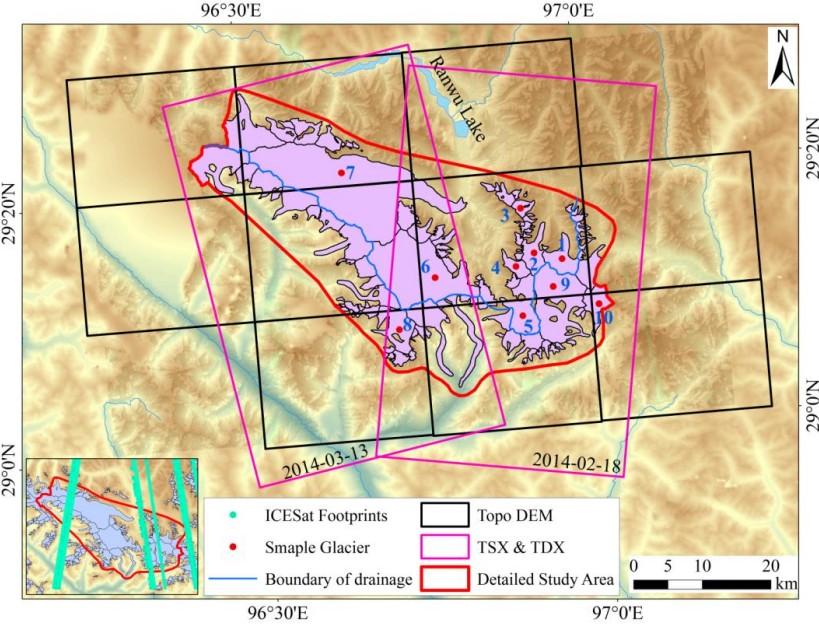

Figure 2. Location of the detailed study area, and distribution of TOPO DEMs, TSX/TDX acquisitions
and ICESat footprints. Numbers indicate specific sample glaciers.



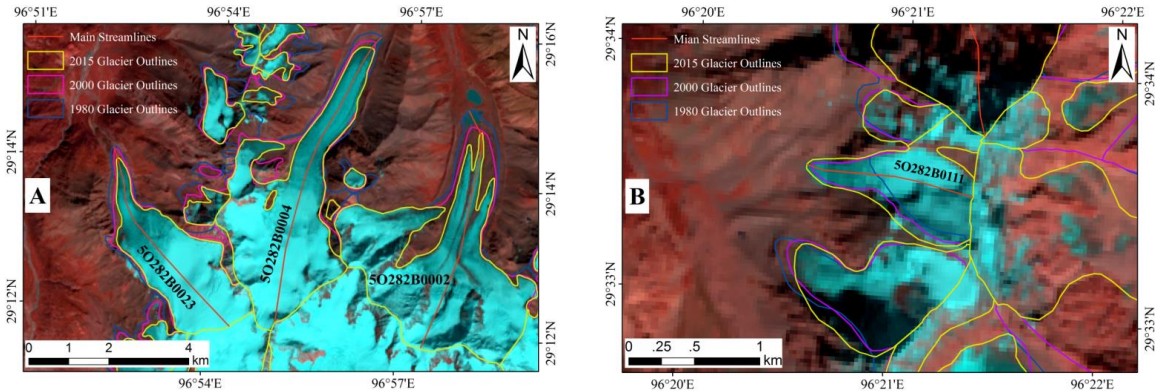

2    Figure 3. Example of glacier outlines derived from imagery collected in 1980, 2000s and 2015. The

3    background image is Landsat OLI image (6 October 2015). (A) Examples of glacier retreat. (B) An

4    example of glacier advance.

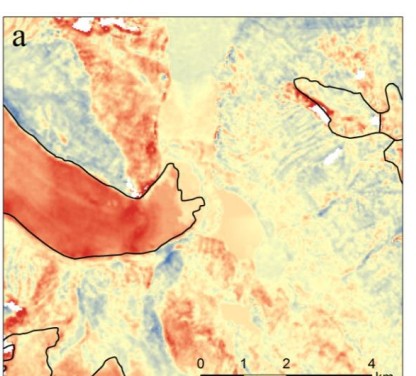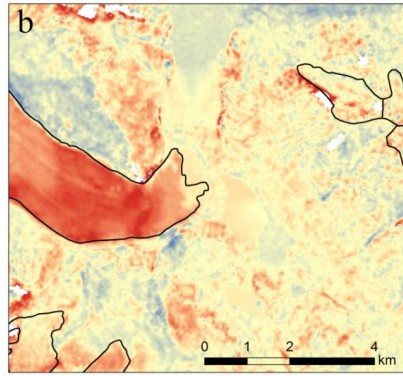

12   Figure 4. Elevation differences estimated between SRTM and TOPO DEM before (a) and after (b) the

13   co-registration in the northern slope of Kangri Karpo Mountain. Location of the data example is shown

14   in Fig. 6A.



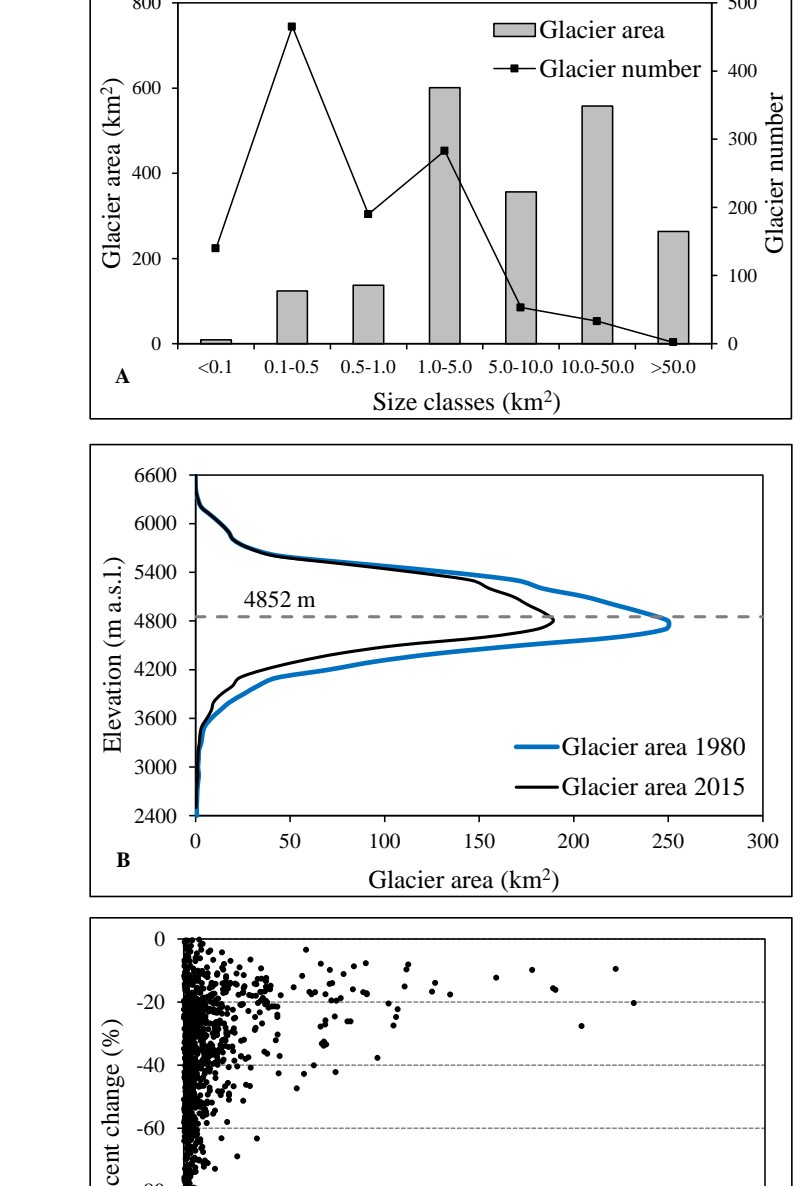

Figure 5. Glacier distribution and change in the Kangri Karpo Mountain. (A) Number and area of glaciers in different size. (B) Percentage changes of glaciers from 1980-2015. (C) Hypsography of glaciers in 1980 and 2015, the dashed line depicts value of median elevation.



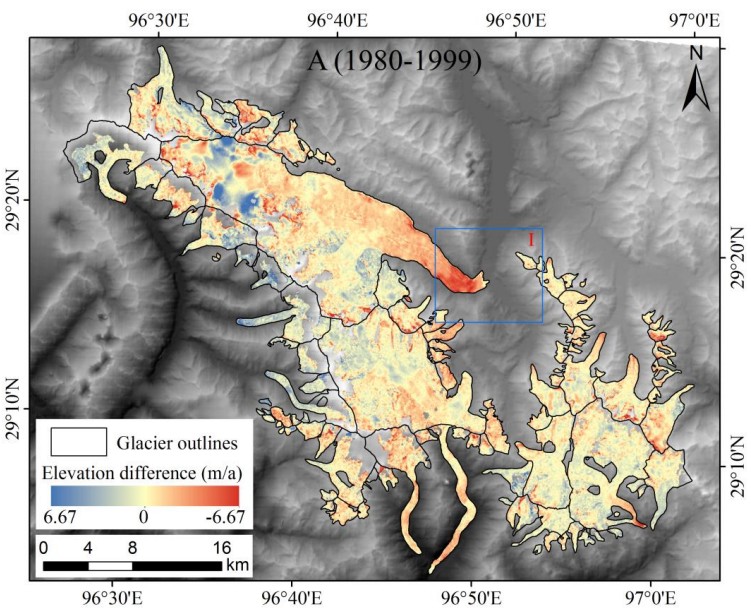

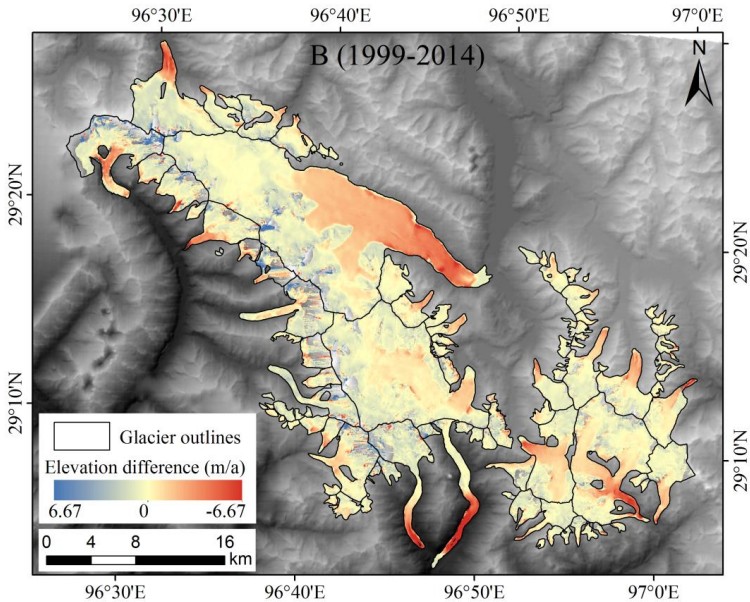



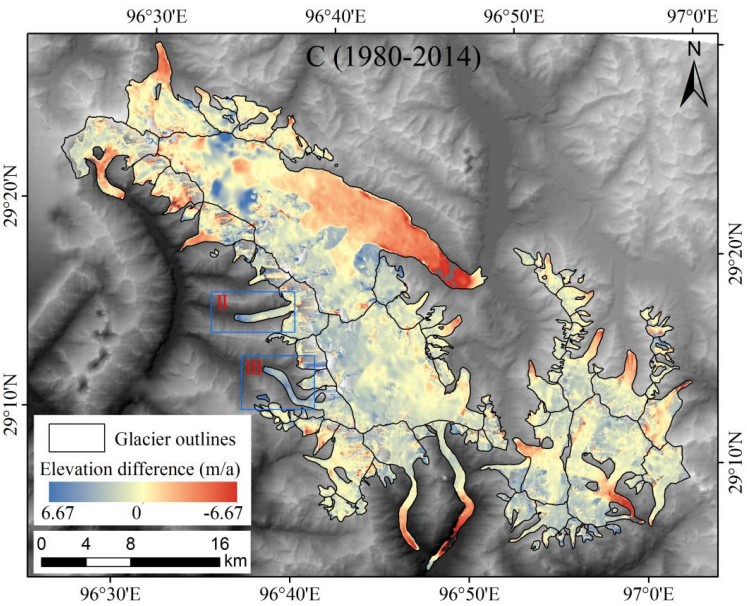

Figure 6. Elevation changes of the detailed study area of Kangri Karpo Mountain from 1980-2014.
The glacier outlines are based on the geometric union of the 1980, 2000s and 2015 glacier extent.
Ⅱ and Ⅲ are two glaciers with positive elevation changes in glacier tongue.

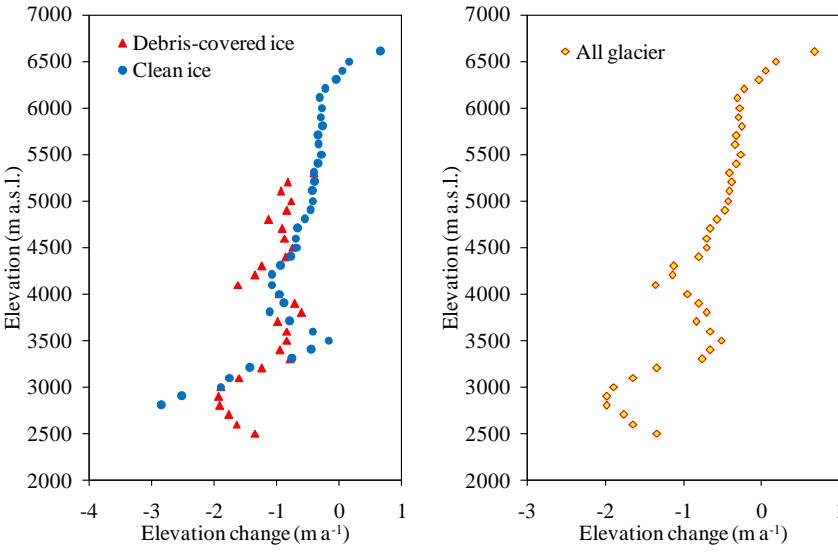

Figure 7. Glacier elevation changes at each 100 m interval by altitude in the detailed study of
Kangri Karpo Mountain for the clean ice, debris-covered ice and all glaciers for the period

9    1980-2014





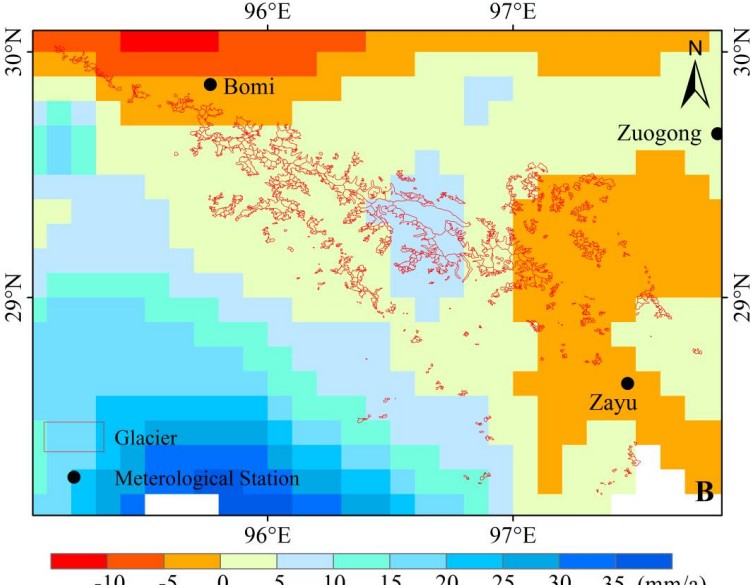

Figure 8. The changes of temperature and precipitation (from May to September) in the Kangri
Karpo Mountain during 1979 - 2012. (A) temperature, (B) precipitation.




Table 1. Overview of satellite images and data sources.

| Date | Source | ID | Pixel size (m) | Utilization |
|---|---|---|---|---|
| October 1980 | Topographic Maps | - | 12 | Glacier identification for 1980 |
| October 1980 | TOPO DEM | H47e016002/H47e017002/ H47e018002 H47e016003/H47e017003/ H47e018003 H47e016004/H47e017004/ H47e018004 H47e016005/H47e017005/ H47e018005 | 30 | Estimation of glacier elevation change |
| 18 December 2001 | Landsat TM | LT51340402001352BJC00 | 30 | Validate and update the GAMDAM and CGI2 inventory |
| 3 January 2002 | Landsat TM | LT51340402002003BJC00 | 30 | |
| 23 October 2001 | Landsat ETM+ | LE71340402001296SGS00 | 15 | |
| 11-22 February 2000 | SRTM C-band | - | 30 | Estimation of glacier elevation change |
| 29 September 2015 | Landsat OLI | LC81330402015272LGN00 | 15 | Glacier identification for 2015 |
| 6 October 2015 | Landsat OLI | LC81340402015279LGN00 | 15 | |
| 25 July 2015 | Landsat OLI | LC81350392015206LGN00 | 15 | |
| 18 February 2014 | TSX/TDX | TDM1_SAR__COS_BIST_SM_S_SRA _20140313T113609_20140313T113617 | 12 | Estimation of glacier elevation change |
| 13 March 2014 | TSX/TDX | TDM1_SAR__COS_BIST_SM_S_SRA _20140313T113609_20140313T113617 | 12 | |

Table 2. Statistics of vertical errors between the TOPO, SRTM and TSX/TDX. MED is mean
elevation difference, STDV is standard deviation, N is the number of considered pixels, SE is
standard error and σ is the overall error of the derived surface elevation change.

| Region | Item | MED (m) | STDV (m) | N | SE (m) | σ (m) |
|---|---|---|---|---|---|---|
| 5O282B basin | SRTM - TOPO | -0.65 | 7.44 | 866 | 0.25 | 0.70 |
| | TSX/TDX - SRTM | -0.90 | 5.83 | 7807 | 0.07 | 0.90 |
| | TSX/TDX - TOPO | -1.42 | 5.07 | 7807 | 0.06 | 1.42 |
| 5O291B basin | SRTM - TOPO | 0.75 | 8.19 | 963 | 0.26 | 0.80 |
| | TSX/TDX - SRTM | 0.07 | 12.68 | 8549 | 0.14 | 0.15 |
| | TSX/TDX - TOPO | 0.71 | 5.50 | 8549 | 0.06 | 0.71 |
| Total | SRTM - TOPO | 0.67 | 16.41 | 1829 | 0.38 | 0.77 |
| | TSX/TDX - SRTM | -0.42 | 9.93 | 16356 | 0.08 | 0.43 |
| | TSX/TDX - TOPO | -0.53 | 5.36 | 16356 | 0.04 | 0.53 |

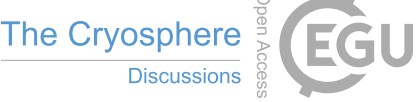

Table 3. Glacier area changes in the Kangri Karpo Mountain from 1980-2015

| Year | 5O291B basin | | 5O291B basin | | Detailed study area | | Whole Mountain range | |
|---|---|---|---|---|---|---|---|---|
| | Area (km²) | Change (% a⁻¹) | Area (km²) | Change (% a⁻¹) | Area (km²) | Change (% a⁻¹) | Area (km²) | Change (% a⁻¹) |
| 1980 | 470.31±4.82 | - | 314.28±4.39 | - | 784.60±5.02 | - | 2728.00±34.24 | - |
| 2000s | 432.91±6.31 | -0.40±0.08 | 287.97±6.02 | -0.42±0.12 | 720.88±7.20 | -0.41±0.06 | - | - |
| 2015 | 406.67±6.76 | -0.40±0.14 | 254.46±7.25 | -0.78±0.22 | 664.88±7.83 | -0.52±0.10 | 2048.50±48.65 | - |
| 1980-2015 | | -0.39±0.05 | | -0.54±0.08 | | -0.44±0.03 | | -0.71±0.06 |





Table 4. Area changes for ten sample glaciers in the detailed study area of Kangri Karpo Mountain.

| ID | Glacier | WGI ID/ GLIMS ID | 1980 Area (km²) | 1980 - 2000 Δa abs. (km²) | 1980 - 2000 Δa rel. (km²) | 1980 - 2000 Rate (% a⁻¹) | 2000 - 2015 Δa abs. (km²) | 2000 - 2015 Δa rel. (km²) | 2000 - 2015 Rate (% a⁻¹) | 1980 - 2015 Δa abs. (km²) | 1980 - 2015 Δa rel. (km²) | 1980 - 2015 Rate (% a⁻¹) |
|---|---|---|---|---|---|---|---|---|---|---|---|---|
| 1 | Danong | 5O282B0002/ G096960E29217N | 15.46 | -0.87 | -5.6% | -0.28% | -1.74 | -11.9% | -0.80% | -2.61 | -16.9% | -0.48% |
| 2 | Parlung NO. 4 | 5O282B0004/ G096920E29228N | 13.52 | -1.56 | -11.5% | -0.58% | -0.97 | -8.1% | -0.54% | -2.53 | -18.7% | -0.53% |
| 3 | Parlung NO. 10 | 5O282B0010/ G096904E29286N | 4.98 | -0.55 | -11.1% | -0.55% | -0.49 | -11.1% | -0.74% | -1.04 | -20.9% | -0.60% |
| 4 | Zuoqiupu | 5O282B0023/ G096891E29212N | 7.46 | -0.76 | -10.2% | -0.51% | -0.43 | -6.4% | -0.43% | -1.19 | -15.9% | -0.45% |
| 5 | Bimaque | 5O282B0025/ G096897E29157N | 26.71 | -1.25 | -4.7% | -0.23% | -2.45 | -9.6% | -0.64% | -3.70 | -13.9% | -0.40% |
| 6 | Xirinongpu | 5O282B0028/ G096745E29216N | 98.99 | -2.50 | -2.5% | -0.13% | -6.21 | -6.4% | -0.43% | -8.71 | -8.8% | -0.25% |
| 7 | Yalong | 5O282B0037/ G096657E29334N | 193.43 | -13.27 | -6.9% | -0.34% | -7.16 | -4.0% | -0.27% | -20.43 | -10.6% | -0.30% |
| 8 | / | 5O291B0151/ G096711E29143N | 19.17 | -0.42 | -2.2% | -0.11% | -1.43 | -7.6% | -0.51% | -1.85 | -9.6% | -0.28% |
| 9 | / | 5O291B0196/ G096943E29175N | 56.45 | -2.42 | -4.3% | -0.21% | -6.11 | -11.3% | -0.75% | -8.54 | -15.1% | -0.43% |
| 10 | / | 5O291B0200/ G097005E29155N | 14.66 | -0.31 | -2.1% | -0.10% | -0.95 | -6.6% | -0.44% | -1.26 | -8.6% | -0.25% |





Table 5. The length change of advanced glaciers in the Gangrigabu Range. The uncertainty of glacier length in 1980 and 2015 are 6 m and 7.5 m, and the uncertainty of length change is 0.27 m a$^{-1}$.

| WGI ID | 1980 | | 2015 | | Length change (m a$^{-1}$) | Lowering of terminate elevation (m) |
|---|---|---|---|---|---|---|
| | Length (m) | Terminate elevation (m) | Length (m) | Terminate elevation (m) | | |
| 5O282B0111 | 762.75 | 5270 | 1300.87 | 4951 | 15.37 | 319 |
| 5O282B0223 | 961.93 | 4884 | 1317.09 | 4637 | 10.15 | 247 |
| 5O282B0225 | 1244.88 | 4705 | 1793.16 | 4483 | 15.67 | 222 |
| 5O282B0226 | 301.13 | 4870 | 648.43 | 4680 | 9.92 | 190 |
| 5O282B0278 | 604.73 | 4876 | 707.97 | 4825 | 2.95 | 51 |
| 5O283A0004 | 1067.55 | 4361 | 2614.65 | 3949 | 44.20 | 412 |
| 5O283B0022 | 481.76 | 4743 | 625.38 | 4624 | 4.10 | 119 |
| 5O291A0004 | 342.07 | 4796 | 798.15 | 4762 | 13.03 | 34 |
| 5O291B0201 | 4045.77 | 3931 | 5047.28 | 3833 | 28.61 | 98 |
| 5O291B0288 | 1277.50 | 4690 | 1898.78 | 4563 | 17.75 | 127 |





Table 6. The length change of glaciers in the detailed study area of Kangri Karpo Mountain. The uncertainty of glacier length in 1980, 2000s and 2015 are 6 m, 7.5 m and 7.5 m, respectively. And the uncertainty of length change during 1980-2000s, 2000s-2015 and 1980-2015 are 0.48 m a$^{-1}$, 0.71 m a$^{-1}$ and 0.27 m a$^{-1}$, respectively.

| WGI ID | Glacier length (m) | | | Length change (m a$^{-1}$) | | |
|---|---|---|---|---|---|---|
| | 1980 | 2000s | 2015 | 1980-2000s | 2000s-2015 | 1980-2015 |
| 5O282B0002 | 6271.16 | 5773.04 | 5635.20 | 24.91 | 9.19 | 18.17 |
| 5O282B0004 | 7756.96 | 7540.24 | 7375.95 | 10.84 | 10.95 | 10.89 |
| 5O282B0010 | 3167.28 | 2970.05 | 2853.12 | 9.86 | 7.80 | 8.98 |
| 5O282B0013 | 3602.30 | 3119.71 | 2960.66 | 24.13 | 10.60 | 18.33 |
| 5O282B0017 | 1631.20 | 1394.38 | 1261.97 | 11.84 | 8.83 | 10.55 |
| 5O282B0023 | 5517.39 | 5431.23 | 5209.88 | 4.31 | 14.76 | 8.79 |
| 5O282B0025 | 5357.49 | 4834.03 | 4548.90 | 26.17 | 19.01 | 23.10 |
| 5O282B0028 | 16890.03 | 16228.66 | 15817.24 | 33.07 | 27.43 | 30.65 |
| 5O282B0034 | 3925.35 | 3860.98 | 3832.31 | 3.22 | 1.91 | 2.66 |
| 5O282B0037 | 32868.46 | 31309.45 | 31105.27 | 77.95 | 13.61 | 50.38 |
| 5O282B0081 | 5306.99 | 5212.33 | 4920.24 | 4.73 | 19.47 | 11.05 |
| 5O282B0083 | 8258.42 | 8102.68 | 7921.12 | 7.79 | 12.10 | 9.64 |
| 5O291B0104 | 8209.80 | 8075.25 | 7922.90 | 6.73 | 10.16 | 8.20 |
| 5O291B0108 | 7570.99 | 7200.65 | 6725.48 | 18.52 | 31.68 | 24.16 |
| 5O291B0113 | 7677.91 | 7627.05 | 7580.98 | 2.54 | 3.07 | 2.77 |
| 5O291B0117 | 15664.48 | 15572.74 | 15456.43 | 4.59 | 7.75 | 5.94 |
| 5O291B0150 | 3509.59 | 2677.17 | 2535.09 | 41.62 | 9.47 | 27.84 |
| 5O291B0151 | 6681.68 | 6329.14 | 6309.40 | 17.63 | 1.32 | 10.64 |
| 5O291B0179 | 13104.49 | 13037.72 | 12473.61 | 3.34 | 37.61 | 18.02 |
| 5O291B0181 | 15536.55 | 15309.66 | 13137.82 | 11.34 | 144.79 | 68.54 |
| 5O291B0196 | 9241.01 | 7157.37 | 6812.94 | 104.18 | 22.96 | 69.37 |
| 5O291B0200 | 7698.33 | 7449.66 | 7013.83 | 12.43 | 29.05 | 19.56 |
| 5O291B0372 | 7681.85 | 7236.61 | 6251.62 | 22.26 | 65.67 | 40.86 |



Table 7. Mean surface elevation changes and mass balance for the single glaciers and different regions in the detailed study area of Kangri Karpo Mountain from 1980-2014. Glacier area is the geometric union of the 1980 glacier area, 2000s glacier area and 2015 glacier area. Mean ΔH is mean surface elevation changes and Mass balance is annual mass budgets.

| Region | | Glacier area (km²) | 1980-2000 | | 2000-2014 | | 1980-2014 | |
|---|---|---|---|---|---|---|---|---|
| | | | Mean ΔH (m) | Mass balance (m w.e. a⁻¹) | Mean ΔH (m) | Mass balance (m w.e. a⁻¹) | Mean ΔH (m) | Mass balance (m w.e. a⁻¹) |
| 1 | 5O282B0002 | 15.48 | -11.05±0.70 | -0.44±0.14 | -13.33±0.91 | -0.86±0.22 | -20.66±1.42 | -0.55±0.22 |
| 2 | 5O282B0004 | 13.63 | -7.70±0.70 | -0.29±0.14 | -10.16±0.91 | -0.65±0.22 | -15.17±1.42 | -0.40±0.22 |
| 3 | 5O282B0010 | 4.99 | -10.31±0.70 | -0.41±0.14 | -10.47±0.91 | -0.67±0.22 | -21.44±1.42 | -0.57±0.22 |
| 4 | 5O282B0023 | 7.46 | -6.28±0.70 | -0.23±0.14 | -8.71±0.91 | -0.56±0.22 | -14.14±1.42 | -0.37±0.22 |
| 5 | 5O282B0025 | 26.72 | -4.24±0.70 | -0.13±0.14 | -13.72±0.91 | -0.88±0.22 | -14.52±1.42 | -0.38±0.22 |
| 6 | 5O282B0028 | 98.99 | -5.93±0.70 | -0.21±0.14 | -8.90±0.91 | -0.57±0.22 | -10.99±1.42 | -0.29±0.22 |
| 7 | 5O282B0037 | 193.45 | -9.21±0.70 | -0.36±0.14 | -15.21±0.91 | -0.98±0.22 | -24.51±1.42 | -0.65±0.22 |
| 5O282B basin | | 471.05 | -7.92±0.70 | -0.30±0.14 | 11.85±0.91 | -0.76±0.22 | -19.13±1.42 | -0.51±0.22 |
| 8 | 5O291B0151 | 19.24 | -8.47±0.80 | -0.33±0.16 | -7.66±0.16 | -0.49±0.04 | -18.56±0.72 | -0.49±0.11 |
| 9 | 5O291B0196 | 56.60 | -3.63±0.80 | -0.11±0.16 | -14.33±0.16 | -0.92±0.04 | -15.25±0.72 | -0.40±0.11 |
| 10 | 5O291B0200 | 14.66 | -2.93±0.80 | -0.08±0.16 | -10.49±0.16 | -0.67±0.04 | -14.16±0.72 | -0.37±0.11 |
| 5O291B basin | | 317.22 | -4.14±0.80 | -0.13±0.16 | -9.74±0.16 | -0.63±0.04 | -14.77±0.72 | -0.39±0.11 |
| Accumulation region | | 530.19 | -4.95±0.77 | -0.22±0.16 | -5.69±0.43 | -0.37±0.10 | -12.06±0.54 | -0.32±0.08 |
| Ablation region | | 258.08 | -5.98±0.77 | -0.27±0.16 | -21.00±0.43 | -1.35±0.10 | -27.64±0.54 | -0.73±0.08 |
| Debris-covered region | | 56.87 | -8.87±0.77 | -0.40±0.16 | -27.39±0.43 | -1.76±0.10 | -33.50±0.54 | -0.89±0.08 |
| Clean-ice region | | 731.43 | -5.00±0.77 | -0.23±0.16 | -9.70±0.43 | -0.62±0.10 | -16.22±0.54 | -0.43±0.08 |
| Total | | 788.31 | -5.30±0.77 | -0.24±0.16 | -11.04±0.43 | -0.71±0.10 | -17.46±0.54 | -0.46±0.08 |