# Peer review of "Recent glacier mass balance and area changes in the Kangri Karpo Mountains from DEMs and glacier inventories"

_The Cryosphere, 2017_

## Referee Comment (RC1) · Anonymous Referee #1 · 30 Sep 2017

General comments:

The study of Wu et al utilizes DEMs derived from Topographic Maps, SRTM DEMs and TerraSAR-X/TanDEM-X to investigate recent glacier mass, length, and area changes in the Kangri Karpo Mountains, southeastern Tibetan Plateau, combined with glacier inventories. In this region, glaciers belong to monsoonal temperature type, which are highly sensitive to global climate change, especially temperature rising. It is important to better understanding of these glaciers' response to climate change for revealing regional glacier change and associated impacts. Also, it is important for better understanding the fundamental mechanism of recent accelerated mass loss in the monsoondominated region.

The science of the manuscript is very interesting, but there are some issues that the authors should consider. The followings are the main points which I think have potential to be expanded upon to increase the value of the study. 1) The study found that most glaciers show significant mass loss and shrinkage, while nine glaciers are in advance for the study period. The authors investigated the reason for advance of these glaciers in the section of 6.2, but this discussion is a little bit simple. The glaciers of this region belong to monsoonal temperature type, where previous studies suggested accelerated mass loss (e.g., Yao et al., 2012) and no such phenomena. Hence, if possible, can the authors provide more discussion for this behavior in this region? 2) As discussed in the manuscript, debris-covered glaciers exist in this region. In particular, the authors found that debris-covered areas are much more thinning on average than clean-ice areas. The manuscript did not introduce how to separate the debris-free and debris-covered regions, Can the authors provide this process in the manuscript? 3) "However, previous studies found that glacier ablation on debris-covered regions were greater than on the exposed ice regions" (Lines 16-17 of page 12). The authors should rewrote this sentence. As previous suggested, ice ablation on debris-covered regions is greater than that on the exposed ice regions, when debris thickness is less than critical thickness (Østrem, 1959; Nakawo and Young, 1981, 1982; Mattson et al., 1993; Kayastha et al., 2000). 4) The English of the manuscript is not well. I strongly advise the authors to improve their English in the manuscript.

Specific comments: P1, L18: "change . . ., increase. . .." changes to "changes . . ., and increases. . .." P2, L8: "Mountains" changes to "Mountain". This should be considered in the manuscript. In some places, it is 'Mountains', and other places, it is 'Mountain'. The authors should make standard format in the manuscript. P2, L37: 1) 'extent' should be 'extents'; 2) The region IS located in . . . P3, L7: 'become' should be 'becomes' P3, L16: 'long' should be 'in length' P4, L36-37: it is not necessary to introduce RGI, and the authors can directly use the second Chinese glacier inventory. P8, L33: delete

while P10, L35: For 'km2', '2' is superscript P11, L24: 'Nyainqentanglha Mountian' should be 'Nyainqentanglha Mountians' Figures 1 and 2: can two figures merger one? Figure 4: I cannot catch two figures difference.

---

## Referee Comment (RC2) · Anonymous Referee #2 · 10 Oct 2017

General Comments:

This paper uses remote sensing imagery and digital elevation models to calculate the mass balance of glaciers in the southeast region of the Tibetan Plateau. Assessing the behavior of these glaciers is important to predict downstream impacts on water resources and the likelihood of glacier-related hazards. The region is of particular interest because the glaciers here receive significant monsoon precipitation and do not have a distinctly separated accumulation and ablation season. The authors present quite a wide range of interesting results, but I think they could go further in using these findings to advance our knowledge about the physical drivers of glacier change in the region.

[Figure]

For example, the authors point to the possibility for glacier surging and/or increases in precipitation to explain their observations of advancing glacier (Page 11, Lines 20-21) and thickening at glacier termini (Page 12, Lines 27-28). This should be expanded: is there any evidence of glacier surging in the region, e.g. from glacier morphological features and patterns? Regarding precipitation, the authors analyze a gridded climate product that shows a mix of increase and decrease in precipitation across the region, but they do not link these results to the inferences about advancing/thickening glaciers. This could be explored further by looking at trends in the accumulation area stake mass balance datasets and exploring more precipitation station data, including a consideration of solid versus liquid precipitation. All of this analysis is particularly important because there seems to be a contradiction between the results presented here and those of Liu et al (2006), who observed 40% of glaciers were advancing during 1980 to 2001.

The authors make inferences about climate controls on the regional glacier mass loss on page 12-13 that are poorly supported. The authors show a trend analysis of a single gridded climate product over a time span that does not exactly match the period of the glacier change record. The spatial distribution of temperature and precipitation change is highly variable and shows both warming/cooling and increased/decreased precipitation across the study area. The authors go on to conclude that climate warming is the primary control on the regional mass balance. A more robust approach would be to subsample the gridded climate product to the glacier polygons, subsample the climate product over the time periods being examined in the glacier assessments, and possibly use degree-day or other methods to more directly relate climate to glacier mass balance. Finally, the authors should note that glaciers have a delayed response to climate forcings so that the changes observed in this study are a response to climate conditions both within and prior to the duration of the observations.

Specific Comments:

Page 1, Line 2 and throughout: Replace "Kangri Karpo Mountain" with "Kangri Karpo

[Figure]

Mountains"

Page 2, Line 5: This statement about a lack of previous studies seems to contradict what follows, which is a list of several studies exploring > 20 years of glacier area and mass change in the region.

Page 2, Line 6-9: List the range of years that these studies explored.

Page 2, Lines 16-22: This paragraph is confusing: how was mass deficit established? From the studies in the previous paragraph, or those referred to in this paragraph? The authors refer to the Nyainqentanglha Mountains, but the reader does not yet know that this is a larger mountain range associated with Kangri Karpo Mountains (we learn this on line 36). For the last sentence in the paragraph, also list the years to which these mass changes apply.

Page 3, Lines 1-8: Use the simple present tense to describe general conditions, unless you are referring to a specific year when the weather conditions had these specific characteristics.

Page 3, Lines 10-12: Suggest replacing this sentence with a statement that the mass turnover is large due to the large ablation and accumulation rates in this maritime region.

Page 3, Line 33: Generally, elevations derived from aerial photography over high elevation, snow covered regions have larger uncertainties due to poor contrast in the imagery. Do the authors have any additional information on this?

Page 5, Line 18: Is it problematic to rely on Google Earth imagery, since it is not possible to know the dates of the images?

Page 5, Line 31: I am unable to access the Yao et al reference: what is a "glacier axis concept"?

Page 6, Lines 1-27: This paragraph is very hard to decipher, especially for a non-expert

in SAR processing. Improvement of the grammar should help in this regard.

Page 7, line 12: Specify how this density uncertainty value was chosen.

Page 8, Line 4: Remove "about"

Page 8, Line 8-9: Replace "distribution in different elevation" with "hypsometry". Also, unless statistical tests for normality were carried out, remove "normal distribution".

Page 8, Line 36: Replace "terminal" with "termini".

Page 8, Lines 36-43: It is a bit difficult reading through all of these numbers. Could the authors condense this into a box plot or something similar?

Page 9, Lines 10-18: The authors report both total mass change over the measurement period, and rates of change (i.e. divided by the number of years). This mix of report is confusing. I recommend only reporting rates, especially when making comparisons between two time periods with different durations (1980-2000 and 2000-2014).

Page 9, Lines 18-26: It is not clear where "50282B" and "50291B" are located on the map. Be consistent in terminology: do not mix "mass deficit" with "mass loss".

Page 9, Lines 27-33: This paragraph appears to be referring to elevation changes at specific locations along the glacier? Should this be referencing Figure 7? Otherwise it is not clear where the data are to support these statements. Also, the terminology in this paragraph is confusing. By definition, the ablation area loses mass, and the accumulation area gains mass. In a particular year, there may be no accumulation area due to unfavorable climate conditions, but it is not possible for "ablation area and accumulation area [to] both experience[d] mass loss".

Page 9, Lines 34-37: more information on the percentage of debris cover on individual glaciers and over the entire region would be valuable.

Page 10, Line 29: "shrinkage" generally refers to thinning, but I think the authors intend to say a decrease in glacier area?

Page 10, Lines 29-32: How do these results reconcile with the findings of Liu et al., 2006, who found 40% of glaciers were advancing in the region between 1980-2001?

Page 11, Lines 12-14: These are interesting ideas about what controls variations glacier advance and retreat, but they should be expanded and backed up with more firm evidence. The authors should cite sources for the velocity increase to explain the decrease in advance rate, otherwise this statement should be probably removed.

Page 11, Lines 15-21: I believe the authors intend to use the term "advancing" instead of "advanced"? If so, it was not clear until this paragraph that any glaciers in the study were advancing during the observation period. This should be clarified earlier in the text. Also, replace "terminals" with "termini" and explain what it means for a terminus to be "closed"?

Page 12, Lines 11-12: It is unclear how these different thinning rates for debris versus clean ice were calculated? Were entire glaciers classified as debris covered, or specific elevation bands? If so, what was the threshold of debris required to classify it in the debris-covered category?

---

## Author Comment (AC1) · 15 Nov 2017

Dear referee,

Thank you for your valuable suggestions and I have already revised the article according to your suggestions. The following are a few answers to some questions.

General comments: (1) "The study found that most glaciers show significant mass loss and shrinkage, while nine glaciers are in advance for the study period. The authors investigated the reason for advance of these glaciers in the section of 6.2, but this discussion is a little bit simple. The glaciers of this region belong to monsoonal tem-

perature type, where previous studies suggested accelerated mass loss (e.g., Yao et al., 2012) and no such phenomena. Hence, if possible, can the authors provide more discussion for this behavior in this region?"

Answer: I have already provided more discussion for advanced glaciers, and we found that advance of individual glaciers resulted from the increase of high precipitation. "For advanced glaciers, the mean glacier size is about 0.51 km2 and mean glacier surface slope is about 27.9 °, most glaciers have an S or SW aspect, and mean accumulation area ratio (AAR) is 51. Previous studies also found advanced glaciers in Kangri Karpo Mountains (Liu et al., 2006; Shi et al., 2006). Compared with the CGI2 and GAMDAM glacier inventory, the location of most glacier termini in 2000 are very close to that in 2014, indicated that glacier advanced mainly occurs before 2000. Due to the special geographic location and climate feature, the qualities of Landsat MSS/TM images are too low to identify glacier termini. Fortunately, two Landsat Thematic Mapper (TM) scenes (LT51340401994189BKT00 and LT51340401988301BJC00) with high quality can be employed in this study. Compared the glacier termini that acquired from Landsat scenes, such as Glacier 5O282B0111 (Fig. 3B), glacier advanced mainly occurs before 1988, and glacier retreated continuously after that (Fig. 7). Main reason for this phenomenon is probably that the increase of high precipitation (Shi et al., 2006). Annual precipitation dataset from 1980 to 2012, collected from the three nearest meteorological stations (Bomi, Zuogong and Zayu), indicated that maximum precipitation (1153 mm in 1988) is 1.6 times the minimum precipitation (714 mm in 1981) at Bomi (29°52′N, 95°46′E, 2736 m a.s.l.), maximum precipitation (683 mm in 1987) is 2.3 times the minimum precipitation (302 mm in 1983) at Zuogong (29°40′N, 97°50′E, 3780 m a.s.l.), and maximum precipitation (1091 mm in 1988) is 1.7 times the minimum precipitation (624 mm in 1982) at Zayu (28°39′N, 97°28′E, 2423 m a.s.l.). Supposing that the precipitation fluctuation in high elevation glacier area had been consistent with that at the three nearest meteorological stations, the change of precipitation or glacier accumulation certainly have significant influence on terminus fluctuation of glaciers. Due to the complicated terrains, the accumulation of glaciers
varies greatly, and the response of glacier movement is not quite the same, individual glaciers advanced during 1980 – 1988."

Figure 7. Terminus changes of Glacier 5O282B0111 from 1980 – 2015.

(2) "As discussed in the manuscript, debris-covered glaciers exist in this region. In particular, the authors found that debris-covered areas are much more thinning on average than clean-ice areas. The manuscript did not introduce how to separate the debris-free and debris-covered regions, Can the authors provide this process in the manuscript?"

Answer: Thank you for your suggestion. Actually, I have already introduced how to separate the debris-free and debris-covered regions in the section of 4.1. "A semi-automated approach using the TM3/TM5 band ratio was applied to delineate glacier outlines in 2015 using Landsat OLI images (Bolch et al., 2010b; Paul et al., 2009; Racoviteanu et al., 2009). To ensure that ice patches were larger than 0.01 km2, a 3 by 3 median filter was applied to eliminate isolated pixel (Bolch et al., 2010b; Wu et al., 2016b). Then, the derived glacier polygons are checked manually against images from adjacent years with less or no snow and cloud-free, to discriminate proglacial lakes, seasonal snow, boulders on the glacier and debris-covered ice." Due to a small proportion of debris-covered regions in Whole Mountain Range, the debris-free and debris-covered regions were separated manually using Landsat OLI images.

(3) & (4) "However, previous studies found that glacier ablation on debris-covered regions were greater than on the exposed ice regions" (Lines 16-17 of page 12). The authors should rewrite this sentence. As previous suggested, ice ablation on debris-covered regions is greater than that on the exposed ice regions, when debris thickness is less than critical thickness (Østrem, 1959; Nakawo and Young, 1981, 1982; Mattson et al., 1993; Kayastha et al., 2000). The English of the manuscript is not well. I strongly advise the authors to improve their English in the manuscript.

Answer: Thank you for your valuable suggestions and I have already revised this sentence and improve English in the manuscript according to your suggestions.

Specific Comments: "Figure 1 and 2: Can two figures merger one?" Answer: At first figure 1 and 2 were drawn together, while a lot of information can't be clearly displayed, such as the locations of sample glaciers and drainage basins. Hence, it is better that there have two figures to show the information of whole study area and detailed study area, respectively.

"Figure 4: I cannot catch two figures difference." Answer: The two figures look similar, but there still have difference between before and after the co-registration, especially in off-glacier regions. Before the co-registration, elevation increase and decrease are both obvious in off-glacier regions, and these phenomena are caused by relative horizontal and vertical distortions between two data sets. After the co-registration, elevation changes are gentle in off-glacier regions, indicated that the pre-processed DEMs were acceptable and suitable for the estimation of changes in glacier mass balance.

Best Regards, Wu Kunpeng and other authors

Please also note the supplement to this comment:
https://www.the-cryosphere-discuss.net/tc-2017-153/tc-2017-153-AC1-supplement.pdf

—————————————

[Figure]

**Supplement:**

[revised manuscript text omitted]

According to the first Chinese Glacier Inventory, the Kangri Karpo Mountains contain 1320 glaciers, with a total area and volume of 2655.2 $km^2$ and 260.3 $km^3$, respectively (Mi et al., 2002). Yalong Glacier (CGI code: 5O282B37) is the largest one among these glaciers (191.4 $km^2$ in surface area and 32.5 km in length), while the Ata Glacier (CGI code: 5O291B181), located on the south slope of the Kangri Karpo Mountains, is the glacier with the lowest terminus at 2450 m, 16.7 km in length and 13.75 $km^2$ in area (Liu et al., 2006). Comparison of photographs taken in different time found that tongue position of Ata Glacier, ice volume and glacial surface conditions have changed greatly in the past decades (Yang et al., 2008).

**3    Data**

**3.1  Topographic Maps**

Five topographic maps on 1:100000 scale and fifty topographic maps on 1:50000 scale generated from aerial photos acquired in October 1980 by the Chinese Military Geodetic Service was employed in the present study. Using a seven parameter transformation method, these maps georeferenced into the 1954 Beijing Geodetic Coordinate System (BJ54) with geoid (datum level is Yellow Sea mean sea level at Qingdao Tidal Observatory in 1956) were re-projected into World Geodetic System 1984 (WGS1984)/Earth Gravity Model 1996 (EGM96) (Xu et al., 2013). The contour lines of the maps were manually digitized and then converted into raster DEM (TOPO DEM) with a 30 m grid cell by employing the Thiessen polygon method (Shangguan et al., 2010; Wei et al., 2015b; Zhang et al., 2016a). According to the national photogrametrical standard issued by the Standardization Administration of the People's Republic of China (GB/T12343.1, 2008), the nominal vertical accuracies of these topographic maps were controlled within 3-5 m for the flat (with slopes < 2 °) and hilly areas (with slope 2-6 °) and controlled within 8-14 m for the mountain (with slope 6-25 °) and high mountain areas (with slope >25 °). 
[revised manuscript text omitted]
., 2015). As the base for the glacier axis concept we assume that the main direction of any given glaciers can be defined as a curved line from its highest to its lowest elevation. At first, the outline of given glacier is divided into two curved lines by its highest and its lowest point. Using the two curved lines, the polygon of given glacier is divided into two regions by Euclidean distance. Glacier axis is the common boundary of the two regions, and it can be defined as the glacier centerline. An error estimation of the resulting glacier centerlines was performed that compared the semi-automatically generated results to high resolution aerial imagery at the terminus of glacier

centerlines. A Corona image with a resolution of 4 m and Google Earth[TM] images with a resolution better than 1 m were used to evaluate the accuracy of glacier centerlines. Hence, the uncertainties of glacier centerlines from Topographic Maps and Landsat images are no more than 6 m and 7.5 m, respectively.

**4.3 Glacier elevation changes**

The TerraSAR-X/TanDEM-X acquisitions were processed by differential SAR interferometry (DInSAR) (Neckel et al., 2013a) using GAMMA SAR and interferometric processing software (Werner et al., 2000).

The interferometric phase of the single-pass TerraSAR-X/TanDEM-X interferogram could be described by

$$\Delta_{\emptyset TSX/TDX} = \Delta_{\emptyset orbit} + \Delta_{\emptyset topo} + \Delta_{\emptyset atm} + \Delta_{\emptyset scat} \tag{1}$$

where $\Delta_{\emptyset TSX/TDX}$ is the difference of phases of phases $\emptyset TSX$ and $\emptyset TDX$ simultaneously acquired by TerraSAR-X and TanDEM-X. $\Delta_{\emptyset orbit}$ is the phase difference induced by the different acquisition geometry of the SAR sensors, and $\Delta_{\emptyset topo}$ is the phase difference induced by topography. $\Delta_{\emptyset atm}$ and $\Delta_{\emptyset scat}$ are the phase differences induced by atmospheric conditions and different scattering on the ground. As the data of TerraSAR-X/TanDEM-X were acquired simultaneously, the same atmospheric conditions and scattering are assumed for both SAR antennas, which set $\Delta_{\emptyset atm}$ and $\Delta_{\emptyset scat}$ in Eq. (1) to zero. $\Delta_{\emptyset orbit}$ could be removed from the interferogram by subtracting a simulated flat-earth phase trend (Rosen et al., 2000).

The DInSAR approach can be described by

$$\Delta_{\emptyset diff} = \Delta_{\emptyset TSX/TDX} - \Delta_{\emptyset SRTM-C} \tag{2}$$

where $\Delta_{\emptyset SRTM-C}$ is the interferometric phase of the February 2000 SRTM-C acquisition. Due to the unavailable of the raw interferometric data of the SRTM-C acquisition, $\
[revised manuscript text omitted]

For advanced glaciers, the mean glacier size is about 0.51 km$^2$ and mean glacier surface slope is about 27.9 °, most glaciers have an S or SW aspect, and mean accumulation area ratio (AAR) is 51. Previous studies also found advanced glaciers in Kangri Karpo Mountains (Liu et al., 2006; Shi et al., 2006). Compared with the CGI2 and GAMDAM glacier inventory, the location of most glacier termini in 2000 are very close to that in 2014, indicated that glacier advanced mainly occurs before 2000. Due to the special geographic location and climate feature, the qualities of Landsat MSS/TM images are too low to identify glacier termini. Fortunately, two Landsat TM

scenes (LT51340401994189BKT00 and LT51340401988301BJC00) with high quality can be employed in this study. Compared the glacier termini that acquired from Landsat scenes, such as Glacier 5O282B0111 (Fig. 3B), glacier advanced mainly occurs before 1988, and glacier retreated continuously after that (Fig. 7). Main reason for this phenomenon is probably that the increase of high precipitation (Shi et al., 2006). Annual precipitation dataset from 1980 to 2012, collected from the three nearest meteorological stations (Bomi, Zuogong and Zayu), indicated that maximum precipitation (1153 mm in 1988) is 1.6 times the minimum precipitation (714 mm in 1981) at Bomi (29°52′N, 95°46′E, 2736 m a.s.l.), maximum precipitation (683 mm in 1987) is 2.3 times the minimum precipitation (302 mm in 1983) at Zuogong (29°40′N, 97°50′E, 3780 m a.s.l.), and maximum precipitation (1091 mm in 1988) is 1.7 times the minimum precipitation (624 mm in 1982) at Zayu (28°39′N, 97°28′E, 2423 m a.s.l.). Supposing that the precipitation fluctuation in high elevation glacier area had been consistent with that at the three nearest meteorological stations, the change of precipitation or glacier accumulation certainly have significant influence on terminus fluctuation of glaciers. Due to the complicated terrains, the accumulation of glaciers varies greatly, and the response of glacier movement is not quite the same, individual glaciers advanced during 1980 – 1988.

[revised manuscript text omitted]
 Mountains during 1980 – 2012. The increase of precipitation results in more glacier accumulation, while glaciers have experienced an intense mass deficit in the detailed study area of Kangri Karpo Mountains. It can be concluded that other factors are playing a more important role in glacier mass deficit. For the change of temperature, warming was present in the detailed study area of Kangri Karpo Mountains during 1980 – 2012. And the warming rate on the northern slope of the Kangri Karpo Mountains is larger than that on the southern slope slightly. A small warming rate was present from 1980 – 2000, and increased to large warming rate thereafter. This is consistent with the tendency of glaciers change. Glaciers have experienced an intense area reduction and mass deficit in the Kangri Karpo Mountains, and mean mass deficit in the drainage basin of 5O282B (located on the northern slope of detailed study area) was larger than that in the drainage basin of 5O291B (located on the southern slope of detailed study area) during 1980 – 2014. Meanwhile, 
[revised manuscript text omitted]

9   drainage basins, where located on north slope and south slope of the detailed study area.

[Figure]

2 Figure 3. Example of glacier outlines derived from imagery collected in 1980, 2000s and 2015. The

3 background image is Landsat OLI image (6 October 2015). (A) Examples of glacier retreat. (B) An

4 example of glacier advance.

[Figure]

12 Figure 4. Elevation differences estimated between SRTM and TOPO DEM before (a) and after (b) the

13 co-registration in the northern slope of Kangri Karpo Mountains. Location of the data example is

14 shown in Fig. 6A.

[Figure]

4 Figure 5. Glacier distribution and change in the Kangri Karpo Mountains. (A) Number and area of

5 glaciers in different size. (B) Hypsography of glaciers in 1980 and 2015, the dashed line depicts

6 value of median elevation. (C) Percentage changes of glaciers from 1980 – 2015.

[revised manuscript text omitted]

---

## Author Comment (AC2) · 15 Nov 2017

Dear referee,

Thank you for your valuable suggestions and I have already revised the article according to your suggestions. The following are a few answers to some questions.

General Comments: (1) "The authors point to the possibility for glacier surging and/or increases in precipitation to explain their observations of advancing glacier (Page 11, Lines 20-21) and thickening at glacier termini (Page 12, Lines 27-28). This should be expanded: is there any evidence of glacier surging in the region, e.g. from glacier

morphological features and patterns? Regarding precipitation, the authors analyze a gridded climate product that shows a mix of increase and decrease in precipitation across the region, but they do not link these results to the inferences about advancing/thickening glaciers. This could be explored further by looking at trends in the accumulation area stake mass balance datasets and exploring more precipitation station data, including a consideration of solid versus liquid precipitation. "

Answer: I have already provided more discussion for advanced glaciers, and we found that advance of individual glaciers resulted from the increase of high precipitation. "For advanced glaciers, the mean glacier size is about 0.51 km2 and mean glacier surface slope is about 27.9 °, most glaciers have an S or SW aspect, and mean accumulation area ratio (AAR) is 51. Previous studies also found advanced glaciers in Kangri Karpo Mountains (Liu et al., 2006; Shi et al., 2006). Compared with the CGI2 and GAMDAM glacier inventory, the location of most glacier termini in 2000 are very close to that in 2014, indicated that glacier advanced mainly occurs before 2000. Due to the special geographic location and climate feature, the qualities of Landsat MSS/TM images are too low to identify glacier termini. Fortunately, two Landsat Thematic Mapper (TM) scenes (LT51340401994189BKT00 and LT51340401988301BJC00) with high quality can be employed in this study. Compared the glacier termini that acquired from Landsat scenes, such as Glacier 5O282B0111 (Fig. 3B), glacier advanced mainly occurs before 1988, and glacier retreated continuously after that (Fig. 7). Main reason for this phenomenon is probably that the increase of high precipitation (Shi et al., 2006). Annual precipitation dataset from 1980 to 2012, collected from the three nearest meteorological stations (Bomi, Zuogong and Zayu), indicated that maximum precipitation (1153 mm in 1988) is 1.6 times the minimum precipitation (714 mm in 1981) at Bomi (29°52′N, 95°46′E, 2736 m a.s.l.), maximum precipitation (683 mm in 1987) is 2.3 times the minimum precipitation (302 mm in 1983) at Zuogong (29°40′N, 97°50′E, 3780 m a.s.l.), and maximum precipitation (1091 mm in 1988) is 1.7 times the minimum precipitation (624 mm in 1982) at Zayu (28°39′N, 97°28′E, 2423 m a.s.l.). Supposing that the precipitation fluctuation in high elevation glacier area had
been consistent with that at the three nearest meteorological stations, the change of precipitation or glacier accumulation certainly have significant influence on terminus fluctuation of glaciers. Due to the complicated terrains, the accumulation of glaciers varies greatly, and the response of glacier movement is not quite the same, individual glaciers advanced during 1980 – 1988."

Figure 7. Terminus changes of Glacier 5O282B0111 from 1980 – 2015.

(2) "The authors make inferences about climate controls on the regional glacier mass loss on page 12-13 that are poorly supported."

Answer: I have already provided more discussion for climate controls on the regional glacier mass loss, and we found that climate warming is the primary control on the regional mass balance. "For the rainfall-variation law, a slightly increasing trend was present in the detailed study area of Kangri Karpo Mountains during 1980 – 2012. The increase of precipitation results in more glacier accumulation, while glaciers have experienced an intense mass deficit in the detailed study area of Kangri Karpo Mountains. It can be concluded that other factors are playing a more important role in glacier mass deficit. For the change of temperature, warming was present in the detailed study area of Kangri Karpo Mountains during 1980 – 2012. And the warming rate on the northern slope of the Kangri Karpo Mountains is larger than that on the southern slope slightly. A small warming rate was present from 1980 – 2000, and increased to large warming rate thereafter. This is consistent with the tendency of glaciers change. Glaciers have experienced an intense area reduction and mass deficit in the Kangri Karpo Mountains, and mean mass deficit in the drainage basin of 5O282B (located on the northern slope of detailed study area) was larger than that in the drainage basin of 5O291B (located on the southern slope of detailed study area) during 1980 – 2014. Meanwhile, the rate of glaciers shrinkage and mass loss from 1980 – 2000 were lower than those from 2000 – 2015. Hence, glaciers change in the Kangri Karpo Mountains can be attributed to climate warming. "

Specific Comments: "Page 2, Lines 16-22: This paragraph is confusing: how was mass deficit established?"

Answer: Thank you for your suggestion and I have already revised this paragraph. "Previous studies have agreed that glaciers in the Kangri Karpo Mountains have experienced mass deficit, nevertheless, the results did differ from each other (Gardelle et al., 2013; Gardner et al., 2013; Kääb et al., 2015; Neckel et al., 2014). Using SRTM DEM and SPOT5 DEM (24 November 2011), glaciers experienced a mean thinning of 0.39 ± 0.16 m a-1 in the Kangri Karpo Mountains (Gardelle et al., 2013). Based on ICESat and SRTM, Kääb et al. (2015), Neckel et al. (2014) and Gardner et al. (2013) acquired different results over the Kangri Karpo Mountains, with glacier thickness loss of 1.34 ± 0.29 m a-1, 0.81 ± 0.32 m a-1 and 0.30 ± 0.13 m a-1 during 2003 to 2009, respectively."

"Page 3, Line 33: Generally, elevations derived from aerial photography over high elevation, snow covered regions have larger uncertainties due to poor contrast in the imagery. Do the authors have any additional information on this?"

Answer: More information was added in this paragraph. "According to the national photogrametrical standard issued by the Standardization Administration of the People's Republic of China (GB/T12343.1, 2008), the nominal vertical accuracies of these topographic maps were controlled within 3-5 m for the flat (with slopes < 2°) and hilly areas (with slope 2-6°) and controlled within 8-14 m for the mountain (with slope 6-25°) and high mountain areas (with slope >25°). Since the slopes of the most of the glacierized areas in the Kangri Karpo Mountains were gentle (∼19°), the vertical accuracy of the TOPO DEM is better than 9 m on glaciers."

"Page 5, Line 18: Is it problematic to rely on Google Earth imagery, since it is not possible to know the dates of the images?"

Answer: Google Earth imagery did not be used in this study. The high-resolution images of Google EarthTM were used to validate the accuracy of the glacier delineation

methods in the second Chinese glacier inventory. Google EarthTM images were captured for seven randomly selected regions, where higher-resolution images in Google EarthTM and nearly simultaneous Landsat images are available (Guo et al., 2015). Due to the same data sources and method of glacier inventory between this study and the second Chinese glacier inventory, the study of Guo et al. (2015) was cited to validate the accuracy of the glacier delineation methods in this study.

"Page 5, Line 31: I am unable to access the Yao et al reference: what is a "glacier axis concept"?"

Answer: Thank you for your suggestion and more information about glacier axis concept was added. "In this study, a new strategy based on a glacier axis concept from glacier morphology perspective was applied that requires only glacier outlines and a DEM as input (Yao et al., 2015). As the base for the glacier axis concept we assume that the main direction of any given glaciers can be defined as a curved line from its highest to its lowest elevation. At first, the outline of given glacier is divided into two curved lines by its highest and its lowest point. Using the two curved lines, the polygon of given glacier is divided into two regions by Euclidean distance. Glacier axis is the common boundary of the two regions, and it can be defined as the glacier centerline."

"Page 6, Lines 1-27: This paragraph is very hard to decipher, especially for a non-expert in SAR processing. Improvement of the grammar should help in this regard."

Answer: I have already revised this paragraph, and the SAR processing was introduced in detail. "The TerraSAR-X/TanDEM-X acquisitions were processed by differential SAR interferometry (DInSAR) (Neckel et al., 2013a) using GAMMA SAR and interferometric processing software (Werner et al., 2000). The interferometric phase of the single-pass TerraSAR-X/TanDEM-X interferogram could be described by $\Delta\_(\hat{a}\acute{L}\breve{E}TSX/TDX) = \Delta\_(\hat{a}\acute{L}\breve{E}orbit) + \Delta\_(\hat{a}\acute{L}\breve{E}topo) + \Delta\_(\hat{a}\acute{L}\breve{E}atm) + \Delta\_(\hat{a}\acute{L}\breve{E}scat)$ (1) where $\Delta\_(\hat{a}\acute{L}\breve{E}TSX/TDX)$ is the difference of phases of phases $\hat{a}\acute{L}\breve{E}TSX$ and $\hat{a}\acute{L}\breve{E}TDX$ simultaneously acquired by TerraSAR-X and TanDEM-X. $\Delta\_(\hat{a}\acute{L}\breve{E}orbit)$ is the

phase difference induced by the different acquisition geometry of the SAR sensors, and $\Delta_{(\text{â}\acute{\text{L}}\check{\text{E}}\text{topo})}$ is the phase difference induced by topography. $\Delta_{(\text{â}\acute{\text{L}}\check{\text{E}}\text{atm})}$ and $\Delta_{(\text{â}\acute{\text{L}}\check{\text{E}}\text{scat})}$ are the phase differences induced by atmospheric conditions and different scattering on the ground. As the data of TerraSAR-X/TanDEM-X were acquired simultaneously, the same atmospheric conditions and scattering are assumed for both SAR antennas, which set $\Delta_{(\text{â}\acute{\text{L}}\check{\text{E}}\text{atm})}$ and $\Delta_{(\text{â}\acute{\text{L}}\check{\text{E}}\text{scat})}$ in Eq. (1) to zero. $\Delta_{(\text{â}\acute{\text{L}}\check{\text{E}}\text{orbit})}$ could be removed from the interferogram by subtracting a simulated flat-earth phase trend (Rosen et al., 2000). The DInSAR approach can be described by $\Delta_{(\text{â}\acute{\text{L}}\check{\text{E}}\text{diff})} = \Delta_{(\text{â}\acute{\text{L}}\check{\text{E}}\text{TSX/TDX})} - \Delta_{(\text{â}\acute{\text{L}}\check{\text{E}}\text{SRTM-C})}$ (2) where $\Delta_{(\text{â}\acute{\text{L}}\check{\text{E}}\text{SRTM-C})}$ is the interferometric phase of the February 2000 SRTM-C acquisition. Due to the unavailable of the raw interferometric data of the SRTM-C acquisition, $\Delta_{(\text{â}\acute{\text{L}}\check{\text{E}}\text{SRTM-C})}$ was simulated from SRTM-C DEM data using the satellite geometry and baseline model of the TerraSAR-X/TanDEM-X pass. Therefore the differential phase $\Delta_{(\text{â}\acute{\text{L}}\check{\text{E}}\text{diff})}$ is solely based on changes in $\Delta_{(\text{â}\acute{\text{L}}\check{\text{E}}\text{topo})}$ between data acquisitions (Neckel et al., 2013b). In order to improve the phase-unwrapping procedure and minimize errors, the unfilled finished SRTM C-band DEM were employed in this study. Before generating the differential interferogram, precise horizontal offset registration and fitting between the SRTM C-band DEM and the TerraSAR-X/TanDEM-X acquisitions is necessary. Based on the relation between the map coordinates of the SRTM C-band DEM segment covering the TerraSAR-X/TanDEM-X master file, and the SAR geometry of the respective master file, an initial lookup table was calculated. While the areas of radar shadows and layover in the TerraSAR-X/TanDEM-X interferogram would induce gaps in the lookup table, a method of linear interpolation between the gap edges in each line of the lookup table was used to fill these gaps. The offsets between the master scene and the simulated intensity of the SRTM C-band DEM, were calculated using cross correlation optimization of the simulated SAR images employing GAMMA's offset_pwrm module. The horizontal registration and geocoding lookup table were refined by these offsets. The SRTM C-band DEM was translated from geographic coordinates into SAR coordinates via the refined geocoding lookup table, and conversely, the final difference map

was translated from SAR coordinates into geographic coordinates. Then a differential interferogram was generated by the TerraSAR-X/TanDEM-X interferogram and the simulated phase of the co-registered SRTM C-band DEM. An adaptive filtering approach was used to filter the differential interferogram (Goldstein and Werner, 1998), and then GAMMA's minimum cost flow (MCF) algorithm was employed to unwrap the flattened differential interferogram. According to the computed phase-to-height sensitivity and select ground control points (GCPs) from the respective off-glacier pixel locations in the SRTM C-band DEM, the unwrapped differential phase was converted to absolute differential heights. While, a residual not covered by the baseline refinement would be existed, and can be regarded as a linear trend that estimated by a two dimensional first order polynomial fit in off-glacier regions. The linear trend and a constant vertical offset were removed from the maps of absolute differential heights. Finally, the resulting data sets were translated to a metric cartographic coordinate system with 30 m $\times$ 30 m pixel spacing (Neckel et al., 2013a). The same method of DInSAR was employed to acquire the glacier elevation change from 1980 to 2014 with the data sets of TOPO DEM and TerraSAR-X/TanDEM-X acquisitions."

"Page 7, line 12: Specify how this density uncertainty value was chosen."

Answer: The value of density uncertainty was chosen by previous studies of Gardner et al. (2013) and Neckel et al. (2015).

"Page 8, Lines 36-43: It is a bit difficult reading through all of these numbers. Could the authors condense this into a box plot or something similar?"

Answer: Thank you for your suggestion. There have Table 5 and Table 6 that can introduce the length changes of advanced glaciers and retreated glaciers clearly.

"Page 9, Lines 34-37: more information on the percentage of debris cover on individual glaciers and over the entire region would be valuable."

Answer: Thank you for your suggestion and more information about debris cover was

added in this paragraph. "Prominent thickening (elevation increase) was found on the termini of two glaciers on the southern slope of the Kangri Karpo Mountains (Fig. 6C, WGI: 5O291B0113 and 5O291B0117). There have 3.79 km2 and 3.70 km2 debris-covered areas on the two glaciers, account for 20.6% and 31.4% of individual glacier areas, respectively. Meanwhile, the lengths of debris-covered regions account for 69.4% and 63.3% of individual glacier lengths. Probably the effect of debris cover, the glacier termini of 5O291B0113 and 5O291B0117 remain stable between October 1980 and October 2015."

"Page 10, Lines 29-32: How do these results reconcile with the findings of Liu et al., 2006, who found 40% of glaciers were advancing in the region between 1980 – 2001?"

Answer: The findings of Liu et al. (2006) and this study both resulted from Topographic Maps that generated from aerial photos acquired in October 1980. Glacier inventory in other mountains of western China, resulted from Topographic Maps, has fewer mistakes that fewer glacier outlines are not accurate. In order to improve the accuracy of glacier inventory in this study, aerial photographs was employed to check and revise glacier outlines that resulted from Topographic Maps. Hence, the results of this study are more reliable and more accurate.

"Page 12, Lines 11-12: It is unclear how these different thinning rates for debris versus clean ice were calculated? Were entire glaciers classified as debris covered, or specific elevation bands? If so, what was the threshold of debris required to classify it in the debris-covered category?"

Answer: Thank you for your suggestion. These different thinning rates for debris versus clean ice were calculated in specific elevation bands. "Interestingly is the bigger thinning on the debris-covered region of -0.99 $\pm$ 0.09 m a-1 on average than clean-ice region of -0.89 $\pm$ 0.08 m w.e. a-1 in the 2800 – 5300 m a.s.l. altitude range from 1980 – 2014 (Fig. 8). 2800 m a.s.l. was the lowest altitude of clean-ice region and 5300 m a.s.l. was the highest altitude of debris-covered region."

[Figure]

Best Regards, Wu Kunpeng and other authors

Please also note the supplement to this comment:
https://www.the-cryosphere-discuss.net/tc-2017-153/tc-2017-153-AC2-supplement.pdf

**Supplement:**

[revised manuscript text omitted]

According to the first Chinese Glacier Inventory, the Kangri Karpo Mountains contain 1320 glaciers, with a total area and volume of 2655.2 $km^2$ and 260.3 $km^3$, respectively (Mi et al., 2002). Yalong Glacier (CGI code: 5O282B37) is the largest one among these glaciers (191.4 $km^2$ in surface area and 32.5 km in length), while the Ata Glacier (CGI code: 5O291B181), located on the south slope of the Kangri Karpo Mountains, is the glacier with the lowest terminus at 2450 m, 16.7 km in length and 13.75 $km^2$ in area (Liu et al., 2006). Comparison of photographs taken in different time found that tongue position of Ata Glacier, ice volume and glacial surface conditions have changed greatly in the past decades (Yang et al., 2008).

**3    Data**

**3.1  Topographic Maps**

Five topographic maps on 1:100000 scale and fifty topographic maps on 1:50000 scale generated from aerial photos acquired in October 1980 by the Chinese Military Geodetic Service was employed in the present study. Using a seven parameter transformation method, these maps georeferenced into the 1954 Beijing Geodetic Coordinate System (BJ54) with geoid (datum level is Yellow Sea mean sea level at Qingdao Tidal Observatory in 1956) were re-projected into World Geodetic System 1984 (WGS1984)/Earth Gravity Model 1996 (EGM96) (Xu et al., 2013). The contour lines of the maps were manually digitized and then converted into raster DEM (TOPO DEM) with a 30 m grid cell by employing the Thiessen polygon method (Shangguan et al., 2010; Wei et al., 2015b; Zhang et al., 2016a). According to the national photogrametrical standard issued by the Standardization Administration of the People's Republic of China (GB/T12343.1, 2008), the nominal vertical accuracies of these topographic maps were controlled within 3-5 m for the flat (with slopes < 2 °) and hilly areas (with slope 2-6 °) and controlled within 8-14 m for the mountain (with slope 6-25 °) and high mountain areas (with slope >25 °). 
[revised manuscript text omitted]
., 2015). As the base for the glacier axis concept we assume that the main direction of any given glaciers can be defined as a curved line from its highest to its lowest elevation. At first, the outline of given glacier is divided into two curved lines by its highest and its lowest point. Using the two curved lines, the polygon of given glacier is divided into two regions by Euclidean distance. Glacier axis is the common boundary of the two regions, and it can be defined as the glacier centerline. An error estimation of the resulting glacier centerlines was performed that compared the semi-automatically generated results to high resolution aerial imagery at the terminus of glacier

centerlines. A Corona image with a resolution of 4 m and Google Earth[TM] images with a resolution better than 1 m were used to evaluate the accuracy of glacier centerlines. Hence, the uncertainties of glacier centerlines from Topographic Maps and Landsat images are no more than 6 m and 7.5 m, respectively.

**4.3 Glacier elevation changes**

The TerraSAR-X/TanDEM-X acquisitions were processed by differential SAR interferometry (DInSAR) (Neckel et al., 2013a) using GAMMA SAR and interferometric processing software (Werner et al., 2000).

The interferometric phase of the single-pass TerraSAR-X/TanDEM-X interferogram could be described by

$$\Delta_{\emptyset TSX/TDX} = \Delta_{\emptyset orbit} + \Delta_{\emptyset topo} + \Delta_{\emptyset atm} + \Delta_{\emptyset scat} \tag{1}$$

where $\Delta_{\emptyset TSX/TDX}$ is the difference of phases of phases $\emptyset TSX$ and $\emptyset TDX$ simultaneously acquired by TerraSAR-X and TanDEM-X. $\Delta_{\emptyset orbit}$ is the phase difference induced by the different acquisition geometry of the SAR sensors, and $\Delta_{\emptyset topo}$ is the phase difference induced by topography. $\Delta_{\emptyset atm}$ and $\Delta_{\emptyset scat}$ are the phase differences induced by atmospheric conditions and different scattering on the ground. As the data of TerraSAR-X/TanDEM-X were acquired simultaneously, the same atmospheric conditions and scattering are assumed for both SAR antennas, which set $\Delta_{\emptyset atm}$ and $\Delta_{\emptyset scat}$ in Eq. (1) to zero. $\Delta_{\emptyset orbit}$ could be removed from the interferogram by subtracting a simulated flat-earth phase trend (Rosen et al., 2000).

The DInSAR approach can be described by

$$\Delta_{\emptyset diff} = \Delta_{\emptyset TSX/TDX} - \Delta_{\emptyset SRTM-C} \tag{2}$$

where $\Delta_{\emptyset SRTM-C}$ is the interferometric phase of the February 2000 SRTM-C acquisition. Due to the unavailable of the raw interferometric data of the SRTM-C acquisition, $\
[revised manuscript text omitted]

For advanced glaciers, the mean glacier size is about 0.51 km$^2$ and mean glacier surface slope is about 27.9 °, most glaciers have an S or SW aspect, and mean accumulation area ratio (AAR) is 51. Previous studies also found advanced glaciers in Kangri Karpo Mountains (Liu et al., 2006; Shi et al., 2006). Compared with the CGI2 and GAMDAM glacier inventory, the location of most glacier termini in 2000 are very close to that in 2014, indicated that glacier advanced mainly occurs before 2000. Due to the special geographic location and climate feature, the qualities of Landsat MSS/TM images are too low to identify glacier termini. Fortunately, two Landsat TM

scenes (LT51340401994189BKT00 and LT51340401988301BJC00) with high quality can be employed in this study. Compared the glacier termini that acquired from Landsat scenes, such as Glacier 5O282B0111 (Fig. 3B), glacier advanced mainly occurs before 1988, and glacier retreated continuously after that (Fig. 7). Main reason for this phenomenon is probably that the increase of high precipitation (Shi et al., 2006). Annual precipitation dataset from 1980 to 2012, collected from the three nearest meteorological stations (Bomi, Zuogong and Zayu), indicated that maximum precipitation (1153 mm in 1988) is 1.6 times the minimum precipitation (714 mm in 1981) at Bomi (29°52′N, 95°46′E, 2736 m a.s.l.), maximum precipitation (683 mm in 1987) is 2.3 times the minimum precipitation (302 mm in 1983) at Zuogong (29°40′N, 97°50′E, 3780 m a.s.l.), and maximum precipitation (1091 mm in 1988) is 1.7 times the minimum precipitation (624 mm in 1982) at Zayu (28°39′N, 97°28′E, 2423 m a.s.l.). Supposing that the precipitation fluctuation in high elevation glacier area had been consistent with that at the three nearest meteorological stations, the change of precipitation or glacier accumulation certainly have significant influence on terminus fluctuation of glaciers. Due to the complicated terrains, the accumulation of glaciers varies greatly, and the response of glacier movement is not quite the same, individual glaciers advanced during 1980 – 1988.

[revised manuscript text omitted]
 Mountains during 1980 – 2012. The increase of precipitation results in more glacier accumulation, while glaciers have experienced an intense mass deficit in the detailed study area of Kangri Karpo Mountains. It can be concluded that other factors are playing a more important role in glacier mass deficit. For the change of temperature, warming was present in the detailed study area of Kangri Karpo Mountains during 1980 – 2012. And the warming rate on the northern slope of the Kangri Karpo Mountains is larger than that on the southern slope slightly. A small warming rate was present from 1980 – 2000, and increased to large warming rate thereafter. This is consistent with the tendency of glaciers change. Glaciers have experienced an intense area reduction and mass deficit in the Kangri Karpo Mountains, and mean mass deficit in the drainage basin of 5O282B (located on the northern slope of detailed study area) was larger than that in the drainage basin of 5O291B (located on the southern slope of detailed study area) during 1980 – 2014. Meanwhile, 
[revised manuscript text omitted]

9   drainage basins, where located on north slope and south slope of the detailed study area.

[Figure]

2 Figure 3. Example of glacier outlines derived from imagery collected in 1980, 2000s and 2015. The

3 background image is Landsat OLI image (6 October 2015). (A) Examples of glacier retreat. (B) An

4 example of glacier advance.

[Figure]

12 Figure 4. Elevation differences estimated between SRTM and TOPO DEM before (a) and after (b) the

13 co-registration in the northern slope of Kangri Karpo Mountains. Location of the data example is

14 shown in Fig. 6A.

[Figure]

4 Figure 5. Glacier distribution and change in the Kangri Karpo Mountains. (A) Number and area of

5 glaciers in different size. (B) Hypsography of glaciers in 1980 and 2015, the dashed line depicts

6 value of median elevation. (C) Percentage changes of glaciers from 1980 – 2015.

[revised manuscript text omitted]

---

## Editor Comment (EC1) · B. Smith (Editor) · 23 Nov 2017

I thank the authors for their improvements to the manuscript, and for their responses to the two referees' comments. I recommend that the authors provide an annotated manuscript showing the changes made in response to the referee's comments. Both referees indicated that the manuscript required only minor revisions, so once the revisions in response to their comments are complete (see my own comments on this below) it seems like the manuscript can proceed.

Regarding the authors' response to referee 1. The additional observation of the glacier terminus with the Landsat-5 scenes and the more detailed analysis of the precipitation

appear to provide a partial response to the reviewer's first concern. I would encourage the authors to edit the corresponding text in their revised manuscript carefully for English, and to consider whether the comparison between the minimum and maximum precipitation at the meteorological stations is the best way to answer this question; the comparison between the maximum and the minimum leaves open the question of whether the minimum was exceptionally low or the maximum was exceptionally high; a comparison between the mean precipitation and the maximum might be more informative. The manuscript should also clarify what the authors mean by "increase high precipitation." Should this be "increased high-altitude precipitation?" The last sentence of the response also does not make sense to me, and probably needs to be explained in more words.

I agree with referee 1 that the changes in figure 4 are somewhat difficult to see. I would recommend that the information about the improvement in the registration between the two DEMs be illustrated with a histogram of the elevation differences before and after the registration procedure.

Regarding the authors' response to referee 2: The authors should more completely respond to the referee's concerns about surging glaciers: The referee's comment seems to be a polite suggestion that there may not be any surging glaciers in the region, so the authors' suggestion on page 11, lines 20-21, that the glaciers were advancing because of surges, may not be correct. The referee also recommended that the authors consider whether precipitation measured at stations would represents snow or rain, which I do not see in the authors' response.

The authors' response to the referee 2's comment about the climate controls is a step in the right direction, but the discussion needs to be more quantitative. What is the magnitude of the precipitation change and of the warming trends for each region and time period? What magnitude do the authors expect to be needed to produce a significant mass-balance trend?

---

## Author Comment (AC3) · 27 Nov 2017

Dear Referee #1,

Thank you for your valuable suggestions and I have already revised the article according to your suggestions. The following are a few answers to some questions.

General comments: (1) "The study found that most glaciers show significant mass loss and shrinkage, while nine glaciers are in advance for the study period. The authors investigated the reason for advance of these glaciers in the section of 6.2, but this discussion is a little bit simple. The glaciers of this region belong to monsoonal temperature type, where previous studies suggested accelerated mass loss (e.g., Yao et al., 2012) and no such phenomena. Hence, if possible, can the authors provide more discussion for this behavior in this region?"

Answer: I have already provided more discussion for advanced glaciers, and we found that advance of individual glaciers resulted from the increase of high precipitation. "For advancing glaciers the mean size is about 0.51 km2, mean surface slope about 27.9 °, most have an S or SW aspect, and a mean accumulation area ratio (AAR) of 51. Previous studies also found advancing glaciers in the Kangri Karpo (Liu et al., 2006; Shi et al., 2006). Comparing the CGI2 and GAMDAM inventories, the location of most glacier termini in 2000 are very close to those in 2014, indicating that the advance mainly occured before 2000. Unfortunately, due to location and climatic features, most Landsat MSS/TM image quality was too low to identify the snouts. Fortunately, two Landsat TM scenes (LT51340401994189BKT00 and LT51340401988301BJC00) did have enough quality to be used. Comparing the Landsat image of the terminus of Glacier 5O282B0111 (Fig. 3B), it could be determined that the advance occurred mainly before 1988 after which time the glacier retreated continuously (Fig. 7), and was likely due to increased precipitation in the 1980s (Shi et al., 2006). Annual precipitation data for 1980–2012 from the three nearest meteorological stations (Bomi, Zuogong and Zayu), indicated that the maximum precipitation was 1.3 times the mean precipitaion in the decade (1153 mm in 1988 vs. 891 mm) at Bomi (29°52′N, 95°46′E, 2736 m a.s.l.). At Zuogong (29°40′N, 97°50′E, 3780 m a.s.l.) the maximum precipitation was 1.5 times the mean (683 mm in 1987 vs. 405 mm), while at Zayu (28°39′N, 97°28′E, 2423 m a.s.l.) it was 1.4 times the mean (1091 mm in 1988 vs. 792 mm). Assuming variations in precipitation at the high-elevation glacier areas reflect those of the three nearest meteorological stations, the increased accumulation could certainly have influenced terminus activity. In complex terrain the accumulation distribution varies greatly so the response of glaciers may differ; some individual glaciers did advance between 1980 and 1988."

Figure 7. Terminus changes of Glacier 5O282B0111 from 1980 – 2015.

(2) "As discussed in the manuscript, debris-covered glaciers exist in this region. In particular, the authors found that debris-covered areas are much more thinning on average than clean-ice areas. The manuscript did not introduce how to separate the debris-free and debris-covered regions, Can the authors provide this process in the manuscript?"

Answer: Thank you for your suggestion. Actually, I have already introduced how to separate the debris-free and debris-covered regions in the section of 4.1. "A semi-automated approach, using the TM3/TM5 band ratio, was applied to delineate glacier outlines in 2015 using Landsat OLI images (Bolch et al., 2010b; Paul et al., 2009; Racoviteanu et al., 2009). To ensure that ice patches were larger than 0.01 km2, a 3 × 3 median filter was applied to eliminate isolated pixels (Bolch et al., 2010b; Wu et al., 2016b). The derived glacier polygons were checked manually against images from adjacent years with less or no snow and cloud-free, to discriminate proglacial lakes, seasonal snow, supraglacial boulders and debris-covered ice (Fig. 3)." Due to a small proportion of debris-covered regions in Whole Mountain Range, the debris-free and debris-covered regions were separated manually using Landsat OLI images.

(3) & (4) "However, previous studies found that glacier ablation on debris-covered regions were greater than on the exposed ice regions" (Lines 16-17 of page 12). The authors should rewrite this sentence. As previous suggested, ice ablation on debris-covered regions is greater than that on the exposed ice regions, when debris thickness is less than critical thickness (Østrem, 1959; Nakawo and Young, 1981, 1982; Mattson et al., 1993; Kayastha et al., 2000). The English of the manuscript is not well. I strongly advise the authors to improve their English in the manuscript.

Answer: Thank you for your valuable suggestions and I have already revised this sentence and improve English in the manuscript according to your suggestions.

Specific Comments: "Figure 1 and 2: Can two figures merger one?" Answer: At first

figure 1 and 2 were drawn together, while a lot of information can't be clearly displayed, such as the locations of sample glaciers and drainage basins. Hence, it is better that there have two figures to show the information of whole study area and detailed study area, respectively.

"Figure 4: I cannot catch two figures difference." Answer: The two figures look similar, but there still have difference between before and after the co-registration, especially in off-glacier regions. Before the co-registration, elevation increase and decrease are both obvious in off-glacier regions, and these phenomena are caused by relative horizontal and vertical distortions between two data sets. After co-registration, histogram statistics of the elevation differences for off-glacier regions showed that elevation difference in off-glacier regions concentrated on the mean elevation difference from 4.94 m to 0.67 m. It is concluded that elevation difference in off-glacier regions have stabilized, the pre-processed DEMs were acceptable and suitable for the estimation of changes in glaciers mass balance.

Best Regards, Wu Kunpeng and other authors

Please also note the supplement to this comment:
https://www.the-cryosphere-discuss.net/tc-2017-153/tc-2017-153-AC3-supplement.pdf

---

## Author Comment (AC4) · 27 Nov 2017

Dear Referee #2,

Thank you for your valuable suggestions and I have already revised the article according to your suggestions. The following are a few answers to some questions.

General Comments: (1) "The authors point to the possibility for glacier surging and/or increases in precipitation to explain their observations of advancing glacier (Page 11, Lines 20-21) and thickening at glacier termini (Page 12, Lines 27-28). This should be expanded: is there any evidence of glacier surging in the region, e.g. from glacier

morphological features and patterns? Regarding precipitation, the authors analyze a gridded climate product that shows a mix of increase and decrease in precipitation across the region, but they do not link these results to the inferences about advancing/thickening glaciers. This could be explored further by looking at trends in the accumulation area stake mass balance datasets and exploring more precipitation station data, including a consideration of solid versus liquid precipitation. "

Answer: I have already provided more discussion for advanced glaciers, and we found that advance of individual glaciers resulted from the increase of high precipitation. "For advancing glaciers the mean size is about 0.51 km2, mean surface slope about 27.9 °, most have an S or SW aspect, and a mean accumulation area ratio (AAR) of 51. Previous studies also found advancing glaciers in the Kangri Karpo (Liu et al., 2006; Shi et al., 2006). Comparing the CGI2 and GAMDAM inventories, the location of most glacier termini in 2000 are very close to those in 2014, indicating that the advance mainly occured before 2000. Unfortunately, due to location and climatic features, most Landsat MSS/TM image quality was too low to identify the snouts. Fortunately, two Landsat TM scenes (LT51340401994189BKT00 and LT51340401988301BJC00) did have enough quality to be used. Comparing the Landsat image of the terminus of Glacier 5O282B0111 (Fig. 3B), it could be determined that the advance occurred mainly before 1988 after which time the glacier retreated continuously (Fig. 7), and was likely due to increased precipitation in the 1980s (Shi et al., 2006). Annual precipitation data for 1980–2012 from the three nearest meteorological stations (Bomi, Zuogong and Zayu), indicated that the maximum precipitation was 1.3 times the mean precipitaion in the decade (1153 mm in 1988 vs. 891 mm) at Bomi (29°52′N, 95°46′E, 2736 m a.s.l.). At Zuogong (29°40′N, 97°50′E, 3780 m a.s.l.) the maximum precipitation was 1.5 times the mean (683 mm in 1987 vs. 405 mm), while at Zayu (28°39′N, 97°28′E, 2423 m a.s.l.) it was 1.4 times the mean (1091 mm in 1988 vs. 792 mm). Assuming variations in precipitation at the high-elevation glacier areas reflect those of the three nearest meteorological stations, the increased accumulation could certainly have influenced terminus activity. In complex terrain the accumulation distri-
bution varies greatly so the response of glaciers may differ; some individual glaciers did advance between 1980 and 1988."

Figure 7. Terminus changes of Glacier 5O282B0111 from 1980 – 2015.

(2) "The authors make inferences about climate controls on the regional glacier mass loss on page 12-13 that are poorly supported."

Answer: I have already provided more discussion for climate controls on the regional glacier mass loss, and we found that climate warming is the primary control on the regional mass balance. "Rainfall increased slightly in the Kangri Karpo during 1980–2012. This increase in precipitation resulted in more glacier accumulation yet the glaciers experienced an intense mass deficit. Other factors must be playing a more important role in this deficit. In the case of temperature, warming was present in the Kangri Karpo during 1980–2012. Meteorological sration records indicate that average air temperature increased in the Kangri Karpo Mountains more than 0.2°C per decade (with confidence level <0.05), higher than the rate of warming in global (0.12°C per decade, 1951–2012). The rate of warming on the northern slope is slightly larger than that on the southern slope. Meteorological sration records showed that average air temperature increased at 0.27°C per decade and 0.25°C per decade in Bomi and Zuo-gong station, higher than Zayu station slightly (0.2°C per decade). While a small warming rate was present from 1980–2000 it increased to large warming rate thereafter. This is consistent with how the glaciers have changed. In the Kangri Karpo they have experienced a substantial area reduction and mass deficit. The mean mass deficit in drainage basin 5O282B (on the northern slope) was larger than that in drainage basin 5O291B (on the southern slope) during 1980–2014. Furthermore, the rate of glacier shrinkage and mass loss from 1980–2000 was less than from 2000–2015. Thus, the changes leading to glacier wastage in the Kangri Karpo can be attributed to climate warming. "

Specific Comments: "Page 2, Lines 16-22: This paragraph is confusing: how was mass

deficit established?"

Answer: Thank you for your suggestion and I have already revised this paragraph. "While previous studies agreed that glaciers in the Kangri Karpo were losing mass, the results did differ from each other. Using SRTM and SPOT5 DEMs (24 November 2011), Gardelle et al. (2013) found a mean thinning of $0.39 \pm 0.16$ m a-1, whereas Kääb et al. (2015), Neckel et al. (2014) and Gardner et al. (2013), using ICESat and SRTM, recorded thinning of $1.34 \pm 0.29$ m a-1, $0.81 \pm 0.32$ m a-1 and $0.30 \pm 0.13$ m a-1 from 2003–2009, respectively."

"Page 3, Line 33: Generally, elevations derived from aerial photography over high elevation, snow covered regions have larger uncertainties due to poor contrast in the imagery. Do the authors have any additional information on this?"

Answer: More information was added in this paragraph. "According to the photogrammetric Chinese National Standard (2008) issued by the Standardization Administration of the People's Republic of China, the nominal vertical accuracy of these topographic maps was within 3-5 m for flat and hilly areas (with slopes of $< 2°$ and 2-6°, respectively) and within 8-14 m for the mountainsides and high mountain areas (with slope of 6-25° and $>25°$, respectively). Since the slopes of the most of the glacierized areas in the Kangri Karpo were gentle ($\sim$19°), the vertical accuracy of the TOPO DEM on the glaciers is better than 9 m."

"Page 5, Line 18: Is it problematic to rely on Google Earth imagery, since it is not possible to know the dates of the images?"

Answer: Google Earth imagery did not be used in this study. The high-resolution images of Google EarthTM were used to validate the accuracy of the glacier delineation methods in the second Chinese glacier inventory. Google EarthTM images were captured for seven randomly selected regions, where higher-resolution images in Google EarthTM and nearly simultaneous Landsat images are available (Guo et al., 2015). Due to the same data sources and method of glacier inventory between this study

and the second Chinese glacier inventory, the study of Guo et al. (2015) was cited to validate the accuracy of the glacier delineation methods in this study.

"Page 5, Line 31: I am unable to access the Yao et al reference: what is a "glacier axis concept"?"

Answer: Thank you for your suggestion and more information about glacier axis concept was added. "In this study, a new method, based on an axis concept derived from the glacier's shape, was applied; requiring only the glacier outline and the DEM as input (Yao et al., 2015). The glacier-axis concept assumes the main direction of any given glacier can be defined as a curved line. The glacier outline is divided initially into two curved lines based on its highest and its lowest elevation. Using these, the glacier polygon is then divided by Euclidean distance into two regions. The common boundary of these two regions is the glacier axis or glacier centerline."

"Page 6, Lines 1-27: This paragraph is very hard to decipher, especially for a non-expert in SAR processing. Improvement of the grammar should help in this regard."

Answer: I have already revised this paragraph, and the SAR processing was introduced in detail. "The TerraSAR-X/TanDEM-X acquisitions were processed by differential SAR interferometry (DInSAR) (Neckel et al., 2013) using GAMMA SAR and interferometric processing software (Werner et al., 2000). The interferometric phase of the single-pass TerraSAR-X/TanDEM-X interferogram can be described by $\triangle_{(\hat{a}\acute{L}\breve{E}TSX/TDX)} = \triangle_{(\hat{a}\acute{L}\breve{E}orbit)} + \triangle_{(\hat{a}\acute{L}\breve{E}topo)} + \triangle_{(\hat{a}\acute{L}\breve{E}atm)} + \triangle_{(\hat{a}\acute{L}\breve{E}scat)}$ (1) where $\triangle_{(\hat{a}\acute{L}\breve{E}TSX/TDX)}$ is the phase difference of phases ∅TSX and ∅TDX simultaneously acquired by TerraSAR-X and TanDEM-X, $\triangle_{(\hat{a}\acute{L}\breve{E}orbit)}$ is that from the different acquisition geometry of the SAR sensors, and $\triangle_{(\hat{a}\acute{L}\breve{E}topo)}$ from topography. $\triangle_{(\hat{a}\acute{L}\breve{E}atm)}$ and $\triangle_{(\hat{a}\acute{L}\breve{E}scat)}$ are the phase differences introduced by atmospheric conditions and different scattering on the ground. As the TerraSAR-X/TanDEM-X data were acquired simultaneously, the same atmospheric conditions and scattering could be assumed for both SAR antennas, thus setting $\triangle_{(\hat{a}\acute{L}\breve{E}atm)}$ and $\triangle_{(\hat{a}\acute{L}\breve{E}scat)}$

in Eq. (1) to zero. $\Delta\_(\text{∅orbit})$ was removed from the interferogram by subtracting a simulated flat-earth phase trend (Rosen et al., 2000). The DInSAR approach can be described by $\Delta\_(\text{∅diff}) = \Delta\_(\text{∅TSX/TDX}) - \Delta\_(\text{∅SRTM-C})$ (2) where $\Delta\_(\text{∅SRTM-C})$ is the February 2000 SRTM-C interferometric phase. Lacking the raw interferometric data, $\Delta\_(\text{∅SRTM-C})$ was simulated from SRTM-C DEM data using the satellite geometry and a baseline model of the TerraSAR-X/TanDEM-X pass. Thus the differential phase $\Delta\_(\text{∅diff})$ is based solely on changes in $\Delta\_(\text{∅topo})$ between data acquisitions (Neckel et al., 2013). To improve the phase-unwrapping procedure and minimize errors, the unfilled, finished, SRTM C-band DEM was employed. Before generating the differential interferogram, precise horizontal offset registration and fitting between the SRTM C-band DEM and the TerraSAR-X/TanDEM-X acquisitions is required. Based on the relation between the map coordinates of the SRTM C-band DEM segment covering the TerraSAR-X/TanDEM-X master file, and the SAR geometry of the respective master file, an initial lookup table was calculated. While the areas of radar shadows and layover in the TerraSAR-X/TanDEM-X interferogram would introduce gaps in the lookup table, a method of linear interpolation between the gap edges in each line of the lookup table was used to fill these gaps. The offsets between the master scene and the simulated intensity of the SRTM C-band DEM, were calculated using cross-correlation optimization of the simulated SAR images employing GAMMA's offset_pwrm module. The horizontal registration and geocoding lookup table were refined by these offsets. The SRTM C-band DEM was translated from geographic coordinates into SAR coordinates via the refined geocoding lookup table, and conversely, the final difference map was translated from SAR coordinates into geographic coordinates. Then a differential interferogram was generated by the TerraSAR-X/TanDEM-X interferogram and the simulated phase of the co-registered SRTM C-band DEM. An adaptive filtering approach was used to filter the differential interferogram (Goldstein and Werner, 1998). GAMMA's minimum cost flow (MCF) algorithm was then employed to unwrap the flattened differential interferogram. According to the computed phase-to-height sensitivity and select ground-control points (GCPs) from respective

off-glacier pixel locations in the SRTM C-band DEM, the unwrapped differential phase was converted to absolute differential heights. While, a residual not covered by the baseline refinement would exist it can be regarded as a linear trend estimated by a two-dimensional first-order polynomial fit in off-glacier regions. The linear trend and a constant vertical offset were removed from the maps of absolute differential heights. Finally, the resulting datasets were translated to a metric cartographic coordinate system with 30 m × 30 m pixel spacing (Neckel et al., 2013). The same DInSAR method was employed to acquire the glacier elevation change from 1980–2014 with the data from the TOPO DEM and TerraSAR-X/TanDEM-X acquisitions."

"Page 7, line 12: Specify how this density uncertainty value was chosen."

Answer: The value of density uncertainty was chosen by previous studies of Gardner et al. (2013) and Neckel et al. (2015). "Page 8, Lines 36-43: It is a bit difficult reading through all of these numbers. Could the authors condense this into a box plot or something similar?"

Answer: Thank you for your suggestion. There have Table 5 and Table 6 that can introduce the length changes of advanced glaciers and retreated glaciers clearly.

"Page 9, Lines 34-37: more information on the percentage of debris cover on individual glaciers and over the entire region would be valuable."

Answer: Thank you for your suggestion and more information about debris cover was added in this paragraph. "A marked thickening (elevation increase) was observed at the termini of two glaciers (5O291B0113 and 5O291B0117) on the southern slope of the Kangri Karpo (Fig. 6C). Substantial debris-cover of 3.79 km2 and 3.70 km2, accounts for 20.6% and 31.4% of their individual area and 69.4% and 63.3% of their length. The termini of these glaciers probably remained stable from October 1980 to October 2015 because of this debris cover."

"Page 10, Lines 29-32: How do these results reconcile with the findings of Liu et al.,

2006, who found 40% of glaciers were advancing in the region between 1980 – 2001?"

Answer: The findings of Liu et al. (2006) and this study both resulted from Topographic Maps that generated from aerial photos acquired in October 1980. Glacier inventory in other mountains of western China, resulted from Topographic Maps, has fewer mistakes that fewer glacier outlines are not accurate. In order to improve the accuracy of glacier inventory in this study, aerial photographs was employed to check and revise glacier outlines that resulted from Topographic Maps. Hence, the results of this study are more reliable and more accurate.

"Page 12, Lines 11-12: It is unclear how these different thinning rates for debris versus clean ice were calculated? Were entire glaciers classified as debris covered, or specific elevation bands? If so, what was the threshold of debris required to classify it in the debris-covered category?"

Answer: Thank you for your suggestion. These different thinning rates for debris versus clean ice were calculated in specific elevation bands. "Thinning was noticeably greater on the glacier debris-cover than the white ice in the 2800–5300 m a.s.l. altitude range from 1980–2014 ($-0.99 \pm 0.09$ m a-1 vs. $-0.89 \pm 0.08$ m w.e. a-1) (Fig. 8). Clean-ice extended down to 2800 m a.s.l. whereas 5300 m a.s.l. was the highest altitude of the debris-covered region. The mass-loss patterns on a debris-covered tongue are complicated, with supraglacial lakes, ice cliffs and a heterogeneous debris cover, (Pellicciotti et al., 2015). Although it is generally believed that ablation rates are retarded with a thick debris-cover due to its insulation effect, some previous studies have found that ablation is greater when the debris is less than a critical thickness (Nakawo and Young, 1981; Pu et al., 2003; Ye et al., 2015; Zhang et al., 2011; Zhang et al., 2016a). The situation of debris-covered regions at lower altitudes with higher temperatures, and the development of supraglacial lakes and ice cliffs, likely contributed to the larger mass loss in those regions (Benn et al., 2012; Sakai and Fujita, 2010)."

Best Regards, Wu Kunpeng and other authors

Please also note the supplement to this comment:
https://www.the-cryosphere-discuss.net/tc-2017-153/tc-2017-153-AC4-
supplement.pdf

---

## Author Comment (AC5) · 27 Nov 2017

Dear Professor Benjamin Smith,

Thank you for your valuable suggestions and I have already revised and modified the revision according to your suggestions. The following are a few answers to some questions.

(1) "A comparison between the mean precipitation and the maximum might be more informative."

Answer: I have already revised and compared the maximum precipitation with the

mean precipitation. "Annual precipitation data for 1980–2012 from the three nearest meteorological stations (Bomi, Zuogong and Zayu), indicated that the maximum precipitation was 1.3 times the mean precipitaion in the decade (1153 mm in 1988 vs. 891 mm) at Bomi (29°52′N, 95°46′E, 2736 m a.s.l.). At Zuogong (29°40′N, 97°50′E, 3780 m a.s.l.) the maximum precipitation was 1.5 times the mean (683 mm in 1987 vs. 405 mm), while at Zayu (28°39′N, 97°28′E, 2423 m a.s.l.) it was 1.4 times the mean (1091 mm in 1988 vs. 792 mm). Assuming variations in precipitation at the high-elevation glacier areas reflect those of the three nearest meteorological stations, the increased accumulation could certainly have influenced terminus activity. In complex terrain the accumulation distribution varies greatly so the response of glaciers may differ; some individual glaciers did advance between 1980 and 1988."

(2) "I would recommend that the information about the improvement in the registration between the two DEMs be illustrated with a histogram of the elevation differences before and after the registration procedure."

Answer: Thank you for your suggestion. I have already added a histogram of the elevation differences before and after the registration. "Relative horizontal and vertical distortions between the two datasets, can be corrected with statistical approaches based on the relationship between elevation difference, slope and aspect (Nuth and Kääb, 2011). Elevation differences in off-glacier regions were used to analyze the consistency of the TOPO and SRTM C-band DEMs (Fig. 4). After co-registration, histogram statistics of the elevation differences for off-glacier regions showed that elevation difference in off-glacier regions concentrated on the mean elevation difference from 4.94 m to 0.67 m. It is concluded that elevation difference in off-glacier regions have stabilized, the pre-processed DEMs were acceptable and suitable for the estimation of changes in glaciers mass balance."

Figure 4. Elevation differences estimated between SRTM and TOPO DEMs before (a) and after (b) co-registration, north slope of the Kangri Karpo. Location of the data example is shown in Fig. 6A.

(3) "The referee's comment seems to be a polite suggestion that there may not be any surging glaciers in the region, so the authors' suggestion on page 11, lines 20-21, that the glaciers were advancing because of surges, may not be correct."

Answer: Thank you for your valuable suggestions and I have already revised this sentence. "Overall, negative elevation changes were found for all glaciers except two on the southern slope of the Kangri Karpo (Fig. 6C). Comparing the average changes of these two tongues from 1980–2000 and 2000–2014, positive changes were found between October 1980 and February 2000, and negative changes after 2000. Unfortunately, the situation in the accumulation areas of these glaciers is unknown due to data voids. This activity might be interpreted as the result of higher precipitation (Shi et al., 2006)"

(4) "The authors' response to the referee 2's comment about the climate controls is a step in the right direction, but the discussion needs to be more quantitative."

Answer: Thank you for your valuable suggestions and I have already revised this sentence. "Rainfall increased slightly in the Kangri Karpo during 1980–2012. This increase in precipitation resulted in more glacier accumulation yet the glaciers experienced an intense mass deficit. Other factors must be playing a more important role in this deficit. In the case of temperature, warming was present in the Kangri Karpo during 1980–2012. Meteorological sration records indicate that average air temperature increased in the Kangri Karpo Mountains more than 0.2°C per decade (with confidence level <0.05), higher than the rate of warming in global (0.12°C per decade, 1951–2012). The rate of warming on the northern slope is slightly larger than that on the southern slope. Meteorological sration records showed that average air temperature increased at 0.27°C per decade and 0.25°C per decade in Bomi and Zuogong station, higher than Zayu station slightly (0.2°C per decade). While a small warming rate was present from 1980–2000 it increased to large warming rate thereafter. This is consistent with how the glaciers have changed. In the Kangri Karpo they have experienced a substantial area reduction and mass deficit. The mean mass deficit in drainage basin 5O282B (on

the northern slope) was larger than that in drainage basin 5O291B (on the southern slope) during 1980–2014. Furthermore, the rate of glacier shrinkage and mass loss from 1980–2000 was less than from 2000–2015. Thus, the changes leading to glacier wastage in the Kangri Karpo can be attributed to climate warming."

Best Regards, Wu Kunpeng and other authors

Please also note the supplement to this comment:
https://www.the-cryosphere-discuss.net/tc-2017-153/tc-2017-153-AC5-supplement.pdf
* * *